# Epigenetic clocks moderate the impact of marital status transitions on health in older adults

Meng-Jung Lin *

Department of Sociology, National Taiwan University, Taipei, Taiwan

* mjlinmj@ntu.edu.tw

## Abstract

Chronological age is commonly used to study aging, but biological aging may more accurately reflect cumulative life experiences and psychosocial stressors. This study examines whether epigenetic clocks function as markers of resilience by assessing how marital status transitions are associated with biological aging and health outcomes in later life. Using data from 1,654 non-Hispanic White participants in the Health and Retirement Study, we analyzed thirteen epigenetic clocks derived from DNA methylation profiles. Ordinary least squares and Cox regression models assessed the associations between marital transitions, depressive symptoms, and mortality, adjusting for genetic and social factors. Interaction terms tested whether epigenetic clocks moderated these associations. Results showed limited associations between marital status and epigenetic clocks. Marital status changes were associated with increased depressive symptoms, but were not statistically significantly associated with mortality risk. In the full sample, interactions between epigenetic clocks and marital status change were not statistically significant for depressive symptoms, although a buffering interaction was observed for the Garagnani clock among men. For mortality, significant moderation was observed for the Hannum, Garagnani, and GrimAge clocks, with moderating patterns concentrated among men. Overall, these interaction results suggest that biologically older individuals, particularly men, exhibited greater resilience to these transitions. These findings raise the possibility that epigenetic clocks reflect accumulated life experiences and psychosocial adaptation, potentially including elements of resilience in older adults.

## Introduction

Aging has long fascinated human societies, from Aristotle's philosophical inquiries into the nature of biological deterioration to Emperor Qin Shi Huang's legendary quest for immortality. Across time and cultures, the collective aspiration has not simply been to halt chronological aging but to extend vitality and preserve youthfulness. Although chronological age is conventionally treated as a neutral administrative

**Data availability statement:** The Health and Retirement Study data used in this study are available at the following link: https://hrs.isr.umich.edu/data-products. Specifically, this study merged the RAND HRS Longitudinal File 2020 (V1) (https://hrsdata.isr.umich.edu/data-products/rand-hrs-archived-data-products), HRS Epigenetic Clocks (v1.0) (https://hrsdata.isr.umich.edu/data-products/hrs-epigenetic-clocks; DOI: https://doi.org/10.7826/LPIK1762), HRS Polygenic Score Data (https://hrsdata.isr.umich.edu/data-products/polygenic-score-data-pgs; DOI: https://doi.org/10.7826/JZEL8053), and RAND HRS Family Data (https://hrsdata.isr.umich.edu/data-products/rand-hrs-family-data-2022). To access HRS public-use files, researchers must create an HRS account, complete the required registration information, and log in through the HRS website. For data products requiring supplemental sensitive-data access, researchers must also submit the Sensitive Health Data Order Form (https://hrsdata.isr.umich.edu/data-products/sensitive-health/order-form) together with a completed and signed Sensitive Data Access Use Agreement.

**Funding:** This research was supported by grants from the National Science and Technology Council of Taiwan, grant numbers 113-2628-H-002-016- and 114-2628-H-002-005-. The funders had no role in study design, data collection and analysis, decision to publish, or preparation of the manuscript.

**Competing interests:** The authors have declared that no competing interests exist.

marker, it carries profound biological, psychological, and social meanings [1–3]. Externally, age structures legal entitlements, governs transitions into and out of institutional roles, and shapes interpersonal expectations. Internally, individuals construct their identities in part through the lens of age. While positive stereotypes emphasize wisdom, reliability, and emotional maturity, negative perceptions focus on rigidity, cognitive decline, and physical frailty [4]. These stereotypes not only shape how older adults are treated by others but may also be associated with self-perceptions, with potential implications for health, social integration, and psychological well-being. Recognizing the positive aspects of aging, particularly the accumulation of resilience and adaptive skills, provides a fuller and more balanced understanding of aging.

One key positive trait associated with aging is resilience, broadly defined as the capacity to adapt successfully to adversity [5]. Older adults often draw upon a reservoir of life experiences, coping strategies, and emotional regulation skills that younger individuals have had less opportunity to cultivate [6,7]. Life experiences, both positive and negative, contribute to a form of psychological maturity that supports adaptive functioning under conditions of stress. Consistent with the adage "what doesn't kill you makes you stronger," many older adults demonstrate an enhanced ability to navigate personal, relational, and health-related adversities. Understanding resilience as an outcome of accumulated life experience reframes aging not as an inevitable decline but as a process of growth and adaptation.

Resilience is a contested concept with multiple definitions, including resistance to harm, recovery after an initial disruption, and longer-term adaptation that unfolds under changing constraints. It is also debated whether resilience should be treated as a stable individual attribute or as a dynamic process shaped by social conditions and access to resources [8,9]. In this study, "resilience" refers to a dynamic process that allows for an initial disruption followed by stabilization or recovery, rather than requiring the complete absence of negative effects [10]. Importantly, studies have shown that resilience depends on resources and exposure contexts that are unequally distributed, and life course perspectives highlight cumulative advantage and disadvantage as a process through which material conditions, social support, and chronic stressors accumulate over time and shape later-life vulnerability and recovery [9,11]. In this view, even though resilience is often discussed as an individual strength, scholars have critiqued this language for downplaying the structural constraints and unequal resources that shape who can adapt and recover [8,9]. Over the life course, these unequal resources and constraints accumulate and help to structure the experiences individuals carry into later life. As these resources and constraints accumulate over the life course, they may become biologically embedded, such that epigenetic aging captures cumulative exposure histories that are relevant to later-life vulnerability and recovery [12–14].

Building on this conceptualization, several theoretical frameworks help to contextualize resilience within aging. Resiliency Theory [15] emphasizes the importance of internal and external protective factors, such as coping skills, competence, and supportive relationships, that enable individuals to thrive despite adversity. Psychological resilience has also been conceptualized as fundamentally tied to emotion regulation,

with affect regulation playing a central role in resilient outcomes [16]. Extending these perspectives into gerontology, Feliciano et al. [17] provided a conceptual analysis of resilience in aging, emphasizing how acceptance, adaptation, and enriched coping strategies contribute to successful aging. Collectively, these frameworks underscore that aging is not merely a period of decline but also one of psychological growth, where older adults draw on a lifetime of emotional and social resources to sustain well-being amid life's challenges.

Viewing aging as a multidimensional process offers a more integrated perspective on how individuals adapt to age-related challenges. Recent theoretical frameworks emphasize the biological, psychological, and social dimensions of aging [2,4]. Biological age reflects the physiological condition of the body, psychological age refers to self-perceptions of aging, and social age encompasses culturally constructed roles and expectations tied to chronological milestones. These dimensions interact in complex ways. For example, an individual may have a younger biological age relative to peers but feel socially marginalized due to age-related stereotypes. Conversely, someone with significant health issues may none-theless maintain a strong sense of psychological youth and active social engagement.

A significant development in the biological measurement of aging has been the introduction of epigenetic clocks. These biomarkers estimate biological age by analyzing patterns of DNA methylation across the genome [18]. Although initially conceptualized as purely biological indicators, a growing body of research suggests that epigenetic clocks are sensitive to social environments and life course exposures. Socioeconomic disadvantage, marital status, occupational stress, life-style behaviors, and psychosocial adversity have all been associated with differences in epigenetic aging [12,19–21]. For instance, analyses using data from the National Longitudinal Study of Adolescent to Adult Health (Add Health) showed that educational attainment, income, health behaviors, and stress exposure robustly predicted variation in epigenetic age acceleration across multiple clocks [22]. Their findings further highlight the extent to which these biomarkers reflect both biological and social processes accumulated over the life course.

The plasticity of biological age aligns closely with life course theories that emphasize cumulative exposure, adaptation, and individual agency [14,23]. Biological systems respond dynamically to social and environmental contexts, adjusting to both adversity and support over time. Within this framework, life transitions can serve as turning points that recalibrate health trajectories both psychologically and biologically. Marital experiences are especially relevant because they com-bine relatively stable social states with the possibility of disruptive transitions, both of which can shape stress exposure, resources, and daily routines across the life course.

Marital transitions, in particular, have been extensively examined in relation to health and longevity. Marriage is asso-ciated with better mental health, reduced risk of chronic disease, and longer survival, especially for men [24,25]. In contrast, divorce and widowhood are often linked to increased psychological distress, diminished social and financial resources, and elevated mortality risk [26–31]. These patterns are commonly interpreted through the dual processes of social causation and social selection. Prior work emphasizes that marriage may influence health through social support, pooled resources, and health-related regulation, while healthier and more advantaged individuals are also more likely to marry and remain married, and those in poorer health are more likely to experience marital disruption [24,25,32]. Impor-tantly, this literature distinguishes between enduring states and transitions. Being married versus unmarried can capture relatively stable differences in social integration and resources, whereas becoming divorced or widowed can operate as an acute stressor with consequences that may evolve through longer-term adaptation, with effects shaped by how recently the transition occurred and whether it involved divorce versus widowhood [33,34]. Gender may further condition these associations. A social-control account argues that spousal monitoring of health behavior is one pathway through which marriage may be especially protective for men [35], consistent with evidence of gender differences in psychological adjustment and health consequences following widowhood and other marital disruptions [24,28,30].

Marital transitions are not merely psychosocial events; they also leave biological imprints. Studies have found that spousal loss is associated with physiological dysregulation, increased inflammation, and markers of accelerated aging [36–38]. In the epigenetic aging literature, emerging evidence using HRS data suggests that marital status is associated

with several epigenetic clocks [38]. Related evidence also links partnership disruption, including divorce and widowhood, to positive DNA methylation age acceleration [39]. Along similar lines, work on social loss further suggests that losing a spouse is associated with broad shifts in DNA methylation, pointing to epigenetic correlates of bereavement-related stress [40].

Although some research has begun to examine associations between marital status and epigenetic aging, fewer studies have focused on whether marital status transitions contribute to biological age acceleration when viewed through a life course lens. Marital status change can reflect qualitatively different events, including union dissolution and union formation, and their health consequences are not necessarily symmetric [33,41]. In addition, life course perspectives emphasize that the consequences of union dissolution can depend on timing and recency. Evidence from meta-analyses indicates that mortality risk is most elevated soon after spousal loss and tends to attenuate with longer follow-up [42,43], and meta-regression evidence on divorce/separation similarly suggests stronger associations in the first few years following dissolution [41]. However, in our analytic sample with DNA methylation and polygenic score data, we do not have complete marital history information for all respondents to identify when a union dissolved prior to the observation period, and there are limited union-formation cases during the study window (20 cases). We therefore operationalize transitions using a binary indicator of any marital status change in the main analyses, and we distinguish union dissolution and union formation in supplementary robustness checks. To establish a baseline association and motivate subsequent analyses of resilience, this study begins by testing the following hypothesis:

H1: Individuals who experience marital status change exhibit greater biological age acceleration compared to those who remain in the same marital status.

Psychological resilience has emerged as a key factor in moderating the biological consequences of life stress. Recent empirical research has expanded this understanding by exploring how resilience processes intersect with biological aging. Cumulative stress exposures, including trauma, discrimination, and chronic adversity, have been consistently linked to accelerated epigenetic aging [44,45]. However, psychological resilience factors, particularly emotion regulation and self-control, can buffer these associations. For instance, prior work shows that stress-related epigenetic aging is more pronounced among individuals with poorer emotion regulation, while stronger self-control appears to mitigate stress-related biological dysregulation, consistent with the idea that psychosocial resources can shape the biological embedding of stress [44]. A study using data from the Health and Retirement Study also found that individuals with higher psychological resilience scores experienced slower epigenetic aging across multiple clock measures [46]. These findings underscore the plasticity of biological aging and point to resilience as a promising target for interventions aimed at promoting healthier aging trajectories [47–49].

Although older age is often portrayed as a period of vulnerability, a growing body of research highlights the emergence of psychological strengths across the life course. Emotional regulation, for example, tends to improve with age, as older adults become more adept at sustaining positive affect and downregulating negative emotions [50,51]. This capacity plays a critical role in buffering the effects of stress on health [52]. Building on these findings, researchers have increasingly framed resilience as a psychological resource that slows biological aging [44,46]. However, this preventive framing may not fully capture the complexity of aging in later life. In some cases, an older epigenetic age may not solely indicate vulnerability. It may also reflect the cumulative imprint of life experiences and the adaptive strategies developed to manage them. From this perspective, epigenetic clocks could serve not only as intervention targets but also as indicators of resilience shaped by lived experience. Resilience, in this broader view, encompasses both psychological and biological dimensions. For instance, studies of individuals exposed to early life adversity, such as combat veterans, suggest that hardship can foster developmental maturity and greater resilience in later life [53].

Similarly, research by Hildon et al. [6] shows that older adults often draw on a repertoire of coping strategies accumulated over the life course to manage new challenges. This perspective suggests that biological aging, as

 

measured by epigenetic clocks, may capture not only cumulative burden but also accumulated resilience. Past studies link resilience-related characteristics to epigenetic aging, including evidence that resilience-related traits can buffer stress-related epigenetic aging [44] and that psychological resilience is associated with epigenetic clocks in HRS [46]. Building on this literature, we use epigenetic clocks as moderators to test whether epigenetic aging profiles condition responses to a later-life transition, with depressive symptoms and mortality as the outcomes. In keeping with the process-oriented definition above, resilience in this study is assessed as reduced downstream consequences following marital status change, rather than requiring the complete absence of any adverse response. Accordingly, we hypothesize:

H2: Epigenetically older individuals exhibit greater resilience to marital status change, such that marital status change is more weakly associated with subsequent depressive symptoms and mortality among epigenetically older individuals than among epigenetically younger individuals.

Gender differences are likely to structure these processes. Men and women differ in both the experience and consequences of marital transitions. Men are more vulnerable to social isolation and health decline following marital loss, whereas women, despite often facing greater economic hardship, may benefit from stronger social networks and emotional coping resources [24,28,54]. Widows may reframe solitude as autonomy, while widowers are more likely to experience it as deprivation [54]. These gendered patterns suggest that the buffering effects of resilience, as indexed by epigenetic clocks, may operate differently by gender. Accordingly, the third hypothesis is proposed:

H3: The moderating effects of epigenetic clocks on mental health and mortality following marital status change are stronger among men than among women.

Using data from the Health and Retirement Study, this study examines how marital status transitions are associated with biological aging and whether epigenetic clocks, conceptualized as proxies for accumulated life experiences, moderate the impact of these transitions on depressive symptoms and mortality. In doing so, it explores whether epigenetic clocks function not only as indicators of aging but also as potential markers of accumulated psychosocial adaptation. By integrating biological, psychological, and social frameworks, the analysis contributes to a more nuanced understanding of resilience in later life and highlights the complex, multidimensional nature of aging.

## Methods

### Data source

Data for this study were drawn from the Health and Retirement Study (HRS) (https://hrs.isr.umich.edu/), a longitudinal, nationally representative survey initiated in 1992 by the Institute for Social Research (ISR) at the University of Michigan and sponsored by the National Institute on Aging (NIA) [55]. The HRS tracks the lives of U.S. adults aged 50 and older through biennial interviews, with the primary objective of capturing the social, economic, and health-related factors contributing to retirement and aging. The de-identified data used in this study were accessed in December 2023; at no point did we have access to any information that could identify individual participants.

In addition to survey data, DNA methylation data were collected from a subsample of HRS participants (N = 4,104) who took part in the 2016 Venous Blood Study [56]. During scheduled home visits, trained phlebotomists collected six tubes of blood, including a 10 mL EDTA whole blood tube. The EDTA tubes were shipped to the University of Minnesota laboratory for processing and DNA extraction. DNA methylation was assayed using the Infinium MethylationEPIC BeadChip, resulting in high-quality data for 4,018 participants after quality control procedures.

To incorporate genetic factors as control variables, the study also utilized genetic data from the HRS [57]. Saliva samples were collected from participants during the 2006, 2008, and 2012 survey waves and genotyped using the Illumina HumanOmni2.5 BeadChips at the Center for Inherited Disease Research. Of the more than 19,000 collected samples,

approximately 15,000 passed quality control checks conducted at the Genetics Coordinating Center at the University of Washington.

Additionally, the HRS includes the RAND HRS Family Data [58], which provides harmonized information on respondents' children, parents, and siblings across multiple waves. This information enabled the examination of how social support related to health outcomes through measures such as geographic proximity and financial transfers between children and parents, enhancing the analysis of health outcomes.

By merging DNA methylation data, GWAS data, and detailed survey information, this study integrates epigenetic, genetic, and social measures to test the three research hypotheses.

## Measures

The study considered the following measures:

**Epigenetic clocks.** Previous research has identified genomic regions where methylation changes correlate with chronological age or, more recently, health outcomes associated with aging [56]. Thirteen epigenetic clocks were constructed by the HRS, including nine first-generation clocks trained on chronological age and four second-generation clocks trained on health-related outcomes [20]. These clocks include Horvath 1, Hannum, Lin, Weidner, Vidal-Bralo, Horvath 2, EpiTOC (Yang), Bocklandt, Zhang, Levine-PhenoAge, Lu-GrimAge, and DunedinPACE. DNAm age was calculated as a weighted sum of methylation values at age-associated CpG sites, with weights derived through machine learning models. Table 1 provides details for the thirteen epigenetic clocks. These clocks served as dependent variables in the first set of analyses and as independent variables and moderators in the second set of analyses predicting health outcomes.

**Dependent variables.** Depressive Symptoms: The CESD (Center for Epidemiologic Studies Depression Scale) score was used to assess depressive symptoms. An abridged eight-item version of the CESD was used in HRS, asking respondents whether, during the past week, they felt depressed, felt that everything they did was an effort, experienced restless sleep, felt unable to get going, felt lonely, enjoyed life, felt sad, and were happy. Responses were coded as "yes"

**Table 1. Details of The Thirteen Epigenetic Clocks in HRS.**

| Clock name | First author (year) | Training phenotype | Tissue type(s) | Number of CpG sites | Unit of measurement | Range in HRS |
|---|---|---|---|---|---|---|
| Horvath 1 | Horvath (2013) | Chronological age | 51 tissues/cells | 353 | Years | 23.3–114.5 |
| Hannum | Hannum (2013) | Chronological age | Whole blood | 71 | Years | 25.1–107.8 |
| Levine | Levine (2018) | Phenotypic age | Whole blood | 513 | Years | 26.7–101.7 |
| Horvath 2 | Horvath (2018) | Chronological age | Skin | 391 | Years | 37.0–101.3 |
| Lin | Lin (2015) | Chronological age | Whole blood | 99 | Years | 1.9–133.3 |
| Weidner | Weidner (2014) | Chronological age | Whole blood | 3 | Years | 25.2–148.9 |
| VidalBralo | Vidal-Bralo (2016) | Chronological age | Whole blood | 8 | Years | 36.5–109.9 |
| EpiTOC | Yang (2016) | Chronological age | Whole blood | 385 | Average % DNAm | 0.0–0.2 |
| Zhang | Zhang (2017) | Mortality risk (time-to-death) | Whole blood | 10 | A unit of mortality risk score | −2.5–0.6 |
| Bocklandt | Bocklandt (2011) | Chronological age | Saliva | 2 | Average % DNAm | 0.1–0.9 |
| Garagnani | Garagnani (2012) | Chronological age | Whole blood | 1 | Average % DNAm | 0.4–1.0 |
| GrimAge | Lu (2019) | Mortality risk (time-to-death) | 7 plasma proteins, smoking pack years | 1030 | Years | 42.7–99.6 |
| DunedinPACE | Belsky (2020) | Rate of change across 18 biomarkers | Whole blood | 46 | Rate of aging in years | 0.7–1.5 |

Sources: Crimmins et al. (2020), Schmitz et al. (2022).

or "no." Positive items were reverse-coded, and the items were summed to create the CESD score, with higher scores indicating more depressive symptoms. The second set of analyses used the 2020 wave CESD score as the outcome. For respondents missing CESD in 2020, we used the next available CESD measure in 2022 to reduce case loss. This approach recovered 30 cases.

Mortality: HRS documented respondents' age at death in months. Death dates were collected from Exit Interviews or based on spouse-reported information. The age-at-death variable was calculated by RAND using birth and death dates, measured in months. Because the measurement of epigenetic clocks is essential for testing the study hypotheses, the analytic sample was restricted to individuals with methylation data, with baseline set at their age in months at the beginning of the 2016 interview wave. Observations were right-censored at the end of the 2020 wave.

**Independent variables.** Marital Status: Respondents reported their current marital status as married, married with spouse absent, partnered, separated, divorced, separated/divorced, widowed, or never married. To avoid sparse categories and to facilitate interpretation, marital status was recoded into four categories: married (including partnered), separated/divorced, widowed, and never married. Marital status measured in 2014 was used to predict epigenetic aging, while marital status in 2016 was used to predict later outcomes.

Marital Status Change: Marital status change was conceptualized as a life event to examine resilience in later life. Since epigenetic clocks were measured in 2016, marital status change was assessed by comparing marital status across 2016, 2018, and 2020. For mortality analyses, marital status change was based on comparisons across 2012, 2014, and 2016, given that death could occur between 2016 and 2020. A change in marital status was coded as 1, while no change was coded as 0. Because union formation is rare in this analytic sample, the primary specification uses this two-category change measure. As a robustness check, we also estimated models using a three-category transition measure (no change, union dissolution including widowhood, and union formation), reported in the S1 File Supplementary Excel Tables.

**Control variables.** Control variables included polygenic scores (PGS) for longevity, number of children ever born, age at first birth, and depressive symptoms, as well as measures of health behaviors, educational attainment, highest parental education, total household wealth, retirement status, chronological age, biological sex, family size, number of living siblings, cohort membership, religious affiliation, and population stratification.

A polygenic score (PGS), also known as a polygenic risk score (PRS), summarizes an individual's genetic predisposition to a particular trait or disease, based on the cumulative effect of multiple genetic variants. It is calculated as the weighted sum of risk alleles: $PGS_i = \sum_{j=1}^{n} w_j x_{ij}$ where $i$ indexes individuals, $j$ indexes SNPs, $w$ denotes the weight for each SNP, and $x$ is the number of risk alleles. Polygenic scores were constructed by the HRS [57] using summary statistics from GWAS studies on longevity [59], reproductive behaviors [60], and depressive symptoms [61]. All PGS were standardized to ease interpretation.

The longevity PGS was included to control for genetic predispositions to longer life, which may confound associations between epigenetic age and mortality. PGS for number of children ever born and age at first birth were included to account for potential marriage selection processes, particularly given generational patterns linking fertility and marriage. Depressive symptoms PGS was controlled because it may be related to both the biological and psychological outcomes of interest. Including these genetic controls helped isolate the contribution of environmental and social experiences to observed patterns in epigenetic aging.

Population Stratification: To control for ancestral differences in allele frequencies that could bias genetic associations, the first ten principal components (PCs) from a principal component analysis of genetic ancestry were included in all regression models, as recommended by Price et al. [62].

Health Behaviors: Health behaviors were captured using measures of vigorous physical activity, alcohol consumption, and smoking history. Vigorous physical activity was assessed with the question: "How often do you take

part in sports or activities that are vigorous, such as running, swimming, cycling, aerobics or gym workouts, tennis, or digging with a spade or shovel?" Responses ranged from "every day" to "never." For analysis, responses were dichotomized into frequent (every day or more than once per week = 1) and infrequent (all other responses = 0). Alcohol consumption was assessed by the question "Do you ever drink any alcoholic beverages, such as beer, wine, or liquor?" (coded yes = 1, no = 0). Smoking history was assessed by the question "Have you ever smoked cigarettes?" (coded yes = 1, no = 0).

Cohort Membership: Respondents were grouped into three birth cohorts: older (AHEAD <1924, CODA 1924–1930, HRS 1931–1941), middle (WB 1942–1947), and younger (EBB 1948–1953, MBB 1954–1959, LBB 1960–1965).

## Analytical strategies

Two sets of analyses were conducted to test the study hypotheses. First, ordinary least squares (OLS) regression models were used to examine the effect of marital status on epigenetic aging. Social factors and polygenic scores were included as covariates to account for potential selection effects.

Second, epigenetic clocks were treated as markers of accumulated life experiences and were used to predict depressive symptoms and mortality. OLS models were employed to predict depressive symptoms, while Cox proportional hazards models were used to predict mortality risk. In addition to assessing the main effects of epigenetic clocks on health outcomes, interaction terms between epigenetic clocks and marital status change were included to test moderation hypotheses. Significant interaction effects would suggest that epigenetic clocks may function as a resilience-related resource, moderating the impact of marital status changes on depressive symptoms and mortality.

To assess gender differences as proposed in Hypothesis 3, all analyses were stratified by biological sex, with separate models estimated for men and women. Because the HRS did not measure gender identity in a way that could be used in these analyses, gender differences were operationalized as differences by this binary sex measure.

Given that the analytic samples were modest in size, we used bootstrap resampling with 1,000 replications to obtain empirical confidence intervals for coefficients in the main tables, which provided an additional check on the stability of standard errors and p values in smaller samples.

Finally, because both epigenetic clocks and polygenic scores were developed primarily in European ancestry populations, analyses were restricted to non-Hispanic White respondents to minimize bias from population stratification. Starting with 4,018 respondents who had DNA-methylation data, 2,314 met the ancestry criterion and had valid polygenic scores. Additional exclusions for missing CESD data in 2020 (464), incomplete marital-status history between 2014 and 2020 (94), missing health-behavior data (18), and missing demographic covariates (84) yielded a final analytic sample of 1,654 for the models predicting epigenetic aging and depressive symptoms. The mortality analysis began with the same 4,018 respondents. After removing 275 individuals who lacked duration-to-death or censoring information, 3,743 remained. Subsequent exclusions for missing polygenic scores (1,554), marital-status history (26), health-behavior data (22), and demographic covariates (110) produced a final sample of 2,031 for the Cox proportional hazards models.

We used ChatGPT (OpenAI, GPT-4o) to improve clarity and grammar. All content was reviewed and finalized by the authors.

## Ethics statement

This study was reviewed and approved by the Research Ethics Committee of National Taiwan University (NTU-REC No.: 202401HS013) and was classified as exempt on January 17, 2024. A waiver of informed consent was granted.



## Results

### Descriptive statistics

Table 2 presents descriptive statistics for the variables used in the CESD score analysis. On average, the epigenetic ages of the sample range from approximately 55–70 years, slightly younger than the respondents' mean chronological age of 70.15 years. Women generally exhibited younger epigenetic ages compared to men. The mean CESD score in 2020 was 1.31, with women reporting higher CESD scores than men in both 2016 and 2020. In total, 21.4% of the analytic sample died between the 2016 and 2020 interview waves, with a slightly higher mortality rate among men (23.1%) compared to women (20.1%). As indicated by the dagger markers in Table 2, several baseline characteristics differed significantly by gender, including CESD scores, marital status distributions, and multiple epigenetic clock measures.

Regarding marital status, about 69.8% of respondents reported being married or partnered in 2016, a figure that declined to 63.8% by 2020, primarily due to an increase in widowhood. Approximately 9.8% of respondents experienced a change in marital status between 2016 and 2020. In 2020, around 63.8% of respondents were married or partnered, 11% were separated or divorced, 21.9% were widowed, and 3.3% had never married. Given the rising prevalence of widowhood among older adults, understanding the potential health and mental well-being consequences of losing a spouse is critical. Accordingly, the subsequent analyses examine how marital status is related to biological aging and how individuals' life experiences, as measured by epigenetic clocks, may help buffer the impact of this stressful transition.

### Marital status, polygenic scores, other social factors, and epigenetic aging

Table 3 presents the models examining the associations between marital status and epigenetic aging. Overall, marital status showed limited associations with epigenetic clocks after controlling for polygenic scores and social factors. However, two marital-status contrasts were statistically significant in the fully adjusted models. Compared to being married or partnered, being separated or divorced was associated with a lower Weidner clock estimate (b = −1.942, SE = 0.759, p < .05), while widowhood was associated with a higher DunedinPACE estimate (b = 0.016, SE = 0.007, p < .05).

In gender-stratified models (see S2 Table), widowed women exhibited a statistically significant positive association with DunedinPACE compared to their married or partnered counterparts, while separated or divorced women exhibited a statistically significant negative association with the Weidner clock. Among men, widowers exhibited a statistically significant negative association with the Bocklandt clock relative to married men. Notably, although the coefficient for being widowed was negative in the model predicting the Bocklandt clock among men, prior research indicates that this clock tends to be negatively correlated with other clocks [20]. In addition, pooled OLS models with gender interactions for all covariates indicate that the separated or divorced coefficients differ significantly between men and women across the epigenetic clock models.

Beyond marital status, several social factors were significantly related to biological aging. Health behaviors followed expected patterns across multiple clocks, with vigorous physical activity generally associated with younger epigenetic ages (e.g., GrimAge: b = −0.576, SE = 0.214, p < .01; DunedinPACE: b = −0.011, SE = 0.005, p < .05) and smoking linked to accelerated aging (e.g., Zhang: b = 0.161, SE = 0.020, p < .001; GrimAge: b = 3.078, SE = 0.189, p < .001; DunedinPACE: b = 0.043, SE = 0.004, p < .001). Having ever consumed alcohol showed inconsistent associations with the clocks (e.g., Horvath 1: b = −0.765, SE = 0.330, p < .05; Horvath 2: b = −0.533, SE = 0.224, p < .05; Bocklandt: b = 0.007, SE = 0.003, p < .05), which may reflect that the measure captured lifetime experimentation rather than drinking intensity or frequency. Occasional alcohol use may be linked to broader social engagement and openness to experience, while heavier drinking likely contributes to biological aging.

Parental education emerged as a consistent protective factor across clocks, with higher parental education associated with younger biological age (e.g., Hannum: b = −0.143, SE = 0.047, p < .01; GrimAge: b = −0.085, SE = 0.039, p < .05; EpiTOC: b = −0.001, SE = 0.000, p < .001). Demographically, women exhibited younger biological ages compared to men across most clocks (e.g., Hannum: b = −2.055, SE = 0.267, p < .001; GrimAge: b = −3.244, SE = 0.203, p < .001;



**Table 2. Descriptive Statistics of the Variables (Frequency (%) or Mean (SD)) (HRS).**

| | All | Male | Female |
|---|---|---|---|
| N | 1,654 (100.0%) | 695 (42.0%) | 959 (58.0%) |
| Epigenetic Clocks | | | |
| Horvath 1 | 66.401 (8.983) | 67.196 (8.811)† | 65.825 (9.068)† |
| Hannum | 55.174 (8.626) | 56.441 (8.491)† | 54.255 (8.611)† |
| Levine | 57.418 (9.520) | 58.359 (8.961)† | 56.737 (9.854)† |
| Horvath 2 | 70.199 (8.251) | 70.946 (8.218)† | 69.657 (8.237)† |
| Lin | 59.209 (10.409) | 60.090 (10.155)† | 58.570 (10.549)† |
| Weidner | 67.071 (11.268) | 67.836 (10.966)† | 66.517 (11.456)† |
| Vidal-Bralo | 64.164 (5.834) | 65.314 (5.605)† | 63.331 (5.859)† |
| EpiTOC (Yang) | 0.066 (0.018) | 0.066 (0.017) | 0.067 (0.018) |
| Zhang | −1.103 (0.443) | −0.981 (0.414)† | −1.191 (0.442)† |
| Bocklandt | 0.383 (0.073) | 0.371 (0.075)† | 0.392 (0.070)† |
| Garagnani | 0.719 (0.071) | 0.717 (0.070) | 0.721 (0.071) |
| GrimAge | 67.838 (8.143) | 69.939 (7.877)† | 66.316 (7.996)† |
| DunedinPACE | 1.063 (0.089) | 1.074 (0.089)† | 1.054 (0.087)† |
| CESD Score | | | |
| CESD Score in 2020 | 1.310 (1.919) | 1.121 (1.821)† | 1.446 (1.976)† |
| CESD Score in 2016 | 1.078 (1.766) | 0.860 (1.524)† | 1.236 (1.908)† |
| Death (Total N = 2,031; M: N = 865; F: N = 1,166) | | | |
| No | 1,597 (78.6%) | 665 (76.9%) | 932 (79.9%) |
| Yes | 434 (21.4%) | 200 (23.1%) | 234 (20.1%) |
| Duration | 45.657 (8.746) | 44.805 (9.315) | 46.289 (8.247) |
| Marital Status Change (2016, 18, 20) | | | |
| No | 1,492 (90.2%) | 625 (89.9%) | 867 (90.4%) |
| Yes | 162 (9.8%) | 70 (10.1%) | 92 (9.6%) |
| Marital Status Change (Three category) (2016, 18, 20) | | | |
| No change | 1,492 (90.2%) | 625 (89.9%) | 867 (90.4%) |
| Union dissolution (incl. widowhood) | 142 (8.6%) | 61 (8.8%) | 81 (8.4%) |
| Union formation | 20 (1.2%) | 9 (1.3%) | 11 (1.1%) |
| Marital Status | | | |
| Marital Status in 2020 | † | | |
| Partnered (married/cohabiting/partnered) | 1,056 (63.8%) | 530 (76.3%) | 526 (54.8%) |
| Separated/Divorced | 182 (11.0%) | 64 (9.2%) | 118 (12.3%) |
| Widowed | 362 (21.9%) | 75 (10.8%) | 287 (29.9%) |
| Never Married | 54 (3.3%) | 26 (3.7%) | 28 (2.9%) |
| Marital Status in 2018 | † | | |
| Partnered (married/cohabiting/partnered) | 1,121 (67.8%) | 560 (80.6%) | 561 (58.5%) |
| Separated/Divorced | 184 (11.1%) | 60 (8.6%) | 124 (12.9%) |
| Widowed | 296 (17.9%) | 50 (7.2%) | 246 (25.7%) |
| Never Married | 53 (3.2%) | 25 (3.6%) | 28 (2.9%) |
| Marital Status in 2016 | † | | |
| Partnered (married/cohabiting/partnered) | 1,154 (69.8%) | 573 (82.4%) | 581 (60.6%) |
| Separated/Divorced | 191 (11.5%) | 60 (8.6%) | 131 (13.7%) |
| Widowed | 255 (15.4%) | 36 (5.2%) | 219 (22.8%) |
| Never Married | 54 (3.3%) | 26 (3.7%) | 28 (2.9%) |
| Marital Status in 2014 | † | | |

*(Continued)*



**Table 2.** (Continued)

| | All | Male | Female |
|---|---|---|---|
| **N** | **1,654 (100.0%)** | **695 (42.0%)** | **959 (58.0%)** |
| Partnered (married/cohabiting/partnered) | 1,191 (72.0%) | 583 (83.9%) | 608 (63.4%) |
| Separated/Divorced | 191 (11.5%) | 64 (9.2%) | 127 (13.2%) |
| Widowed | 218 (13.2%) | 23 (3.3%) | 195 (20.3%) |
| Never Married | 54 (3.3%) | 25 (3.6%) | 29 (3.0%) |
| Health Lifestyles | | | |
| Vigorous Physical Activity (2016) | † | | |
| No | 1,178 (71.2%) | 463 (66.6%) | 715 (74.6%) |
| Yes | 476 (28.8%) | 232 (33.4%) | 244 (25.4%) |
| Ever Drinks Any Alcohol (2016) | | | |
| No | 624 (37.7%) | 245 (35.3%) | 379 (39.5%) |
| Yes | 1,030 (62.3%) | 450 (64.7%) | 580 (60.5%) |
| Ever Smokes (2016) | † | | |
| No | 754 (45.6%) | 260 (37.4%) | 494 (51.5%) |
| Yes | 900 (54.4%) | 435 (62.6%) | 465 (48.5%) |
| Vigorous Physical Activity (2014) | † | | |
| No | 1,143 (69.1%) | 449 (64.6%) | 694 (72.4%) |
| Yes | 511 (30.9%) | 246 (35.4%) | 265 (27.6%) |
| Ever Drinks Any Alcohol (2014) | † | | |
| No | 598 (36.2%) | 232 (33.4%) | 366 (38.2%) |
| Yes | 1,056 (63.8%) | 463 (66.6%) | 593 (61.8%) |
| Ever Smokes (2014) | † | | |
| No | 754 (45.6%) | 260 (37.4%) | 494 (51.5%) |
| Yes | 900 (54.4%) | 435 (62.6%) | 465 (48.5%) |
| Polygenic Score | | | |
| Longevity PGS | 0.023 (0.985) | 0.012 (1.004) | 0.031 (0.972) |
| Depressive Symptom PGS | −0.044 (1.005) | −0.050 (0.968) | −0.040 (1.032) |
| Number of Children Ever Born PGS | −0.034 (0.992) | −0.014 (1.015) | −0.048 (0.976) |
| Age at First Birth PGS | 0.055 (0.980) | 0.043 (0.984) | 0.064 (0.977) |
| Socioeconomic Background | | | |
| Years of Education | 13.875 (2.270) | 14.033 (2.374)† | 13.761 (2.185)† |
| Parental Education | 12.116 (3.013) | 12.262 (2.973) | 12.010 (3.040) |
| Total Wealth | | | |
| Total Wealth (2016) | 7.074 (12.837) | 7.346 (12.728) | 6.877 (12.918) |
| Total Wealth (2014) | 6.538 (12.490) | 6.870 (13.371) | 6.297 (11.811) |
| Retirement Status | | | |
| Retirement Status (2016) | † | | |
| Not Retired | 398 (24.1%) | 179 (25.8%) | 219 (22.8%) |
| Completely Retired | 977 (59.1%) | 384 (55.3%) | 593 (61.8%) |
| Partly Retired | 250 (15.1%) | 132 (19.0%) | 118 (12.3%) |
| Question Irrelevant | 29 (1.8%) | 0 (0.0%) | 29 (3.0%) |
| Retirement Status (2014) | † | | |
| Not Retired | 504 (30.5%) | 211 (30.4%) | 293 (30.6%) |
| Completely Retired | 809 (48.9%) | 324 (46.6%) | 485 (50.6%) |
| Partly Retired | 299 (18.1%) | 155 (22.3%) | 144 (15.0%) |

*(Continued)*

**Table 2.** (Continued)

| | All | Male | Female |
|---|---|---|---|
| **N** | **1,654 (100.0%)** | **695 (42.0%)** | **959 (58.0%)** |
| Question Irrelevant | 42 (2.5%) | 5 (0.7%) | 37 (3.9%) |
| Demographic Characteristics | | | |
| Age | | | |
| Age at 2016 | 70.150 (8.896) | 70.279 (8.587) | 70.056 (9.117) |
| Age at 2014 | 67.981 (8.926) | 68.109 (8.609) | 67.887 (9.152) |
| Cohort | | | |
| Old | 608 (36.8%) | 263 (37.8%) | 345 (36.0%) |
| Middle | 327 (19.8%) | 137 (19.7%) | 190 (19.8%) |
| Young | 719 (43.5%) | 295 (42.4%) | 424 (44.2%) |
| Family Size | | | |
| Family Size (2016) | 2.051 (0.886) | 2.129 (0.804)† | 1.994 (0.937)† |
| Family Size (2014) | 2.103 (0.921) | 2.176 (0.800)† | 2.051 (0.997)† |
| Number of Living Siblings | | | |
| Number of Living Siblings (2016) | 2.433 (1.860) | 2.403 (1.823) | 2.456 (1.887) |
| Number of Living Siblings (2014) | 2.446 (1.872) | 2.412 (1.842) | 2.471 (1.895) |
| Religious Affiliation† | | | |
| Protestant | 1,036 (62.6%) | 414 (59.6%) | 622 (64.9%) |
| Catholic | 391 (23.6%) | 160 (23.0%) | 231 (24.1%) |
| None or No Preference | 182 (11.0%) | 100 (14.4%) | 82 (8.6%) |
| Other | 45 (2.7%) | 21 (3.0%) | 24 (2.5%) |

Note: † indicates a statistically significant male–female difference (p<0.05, two-sided test). For continuous variables, † appears next to the Male and Female values when the mean difference between men and women is statistically significant (two-sided t-test). For categorical variables, † appears in the All column on the variable header row to indicate that the overall distribution differs by gender (chi-square omnibus test), rather than differences in any single category.

DunedinPACE: b = −0.018, SE = 0.004, p < .001). Chronological age was positively associated with most epigenetic clocks (e.g., Horvath 1: b = 0.695, SE = 0.043, p < .001; Weidner: b = 0.552, SE = 0.080, p < .001), although it was negatively associated with the Bocklandt clock (b = −0.003, SE = 0.000, p < .001) and unrelated to DunedinPACE (b = −0.000, SE = 0.001, p > .05), consistent with the latter's focus on accelerated biological aging rather than chronological age per se.

Regarding genetic factors, polygenic scores for longevity, number of children ever born, and age at first birth were generally not significantly associated with biological aging. In summary, after accounting for genetic predispositions, social factors such as marital status, health behaviors, and socioeconomic background played important roles in shaping biological aging. These results provide limited support for Hypothesis 1 in the marital-status models, because significant associations were observed for only a subset of clocks and the separated or divorced association was not consistently in the direction of older biological age.

## Epigenetic aging, marital status change, and CESD score

S3 Table presents models examining the main effects of epigenetic clocks and marital status change on CESD scores. Overall, most clocks were not significantly associated with depressive symptoms in the full sample. Two exceptions were Hannum and GrimAge, both of which showed positive associations with CESD, indicating that higher epigenetic age corresponded with higher depressive symptoms (Hannum: b = 0.016, SE = 0.008, p < .05; GrimAge: b = 0.038, SE = 0.012, p < .01). In contrast, marital status change was consistently associated with an increase of approximately 0.61 points in



**Table 3. Ordinary Least Squares Models Using Marital Status, PGSs, and Social Factors to Predict Epigenetic Clocks (HRS).**

| Variables | Model 1 Horvath 1 | Model 2 Hannum | Model 3 Levine | Model 4 Horvath 2 | Model 5 Lin | Model 6 Weidner | Model 7 Vidal-Bralo | Model 8 EpiTOC | Model 9 Zhang | Model 10 Bocklandt | Model 11 Garagnani | Model 12 GrimAge | Model 13 Dunedin-inPACE |
|---|---|---|---|---|---|---|---|---|---|---|---|---|---|
| **2014 Marital Status (Ref.= Married/Partnered)** | | | | | | | | | | | | | |
| Separated/Divorced | 0.269 (0.490) | -0.584 (0.404) | -0.707 (0.557) | -0.191 (0.360) | -0.251 (0.557) | -1.942* (0.759) | 0.075 (0.397) | 0.001 (0.001) | -0.017 (0.031) | -0.004 (0.006) | -0.001 (0.004) | 0.211 (0.315) | 0.002 (0.007) |
| Widowed | -0.705 (0.502) | 0.043 (0.411) | 0.396 (0.564) | -0.138 (0.373) | 0.448 (0.650) | 0.799 (0.890) | 0.529 (0.429) | -0.001 (0.001) | 0.061+ (0.033) | -0.007 (0.006) | -0.004 (0.004) | 0.488 (0.329) | 0.016* (0.007) |
| Never Married | 0.995 (0.815) | 0.148 (0.659) | -0.487 (1.175) | 0.129 (0.498) | 1.668+ (0.977) | -0.669 (1.450) | 0.621 (0.792) | -0.002 (0.002) | -0.033 (0.064) | -0.014 (0.010) | -0.004 (0.007) | 0.577 (0.600) | 0.015 (0.013) |
| **2014 Health Lifestyle** | | | | | | | | | | | | | |
| Vigorous Physical Activity | -0.648+ (0.348) | -0.310 (0.263) | -0.574 (0.353) | -0.342 (0.242) | 0.063 (0.433) | -0.441 (0.567) | -0.315 (0.254) | -0.001 (0.001) | -0.012 (0.022) | -0.000 (0.004) | -0.004 (0.003) | -0.576** (0.214) | -0.011* (0.005) |
| Ever Drinks Any Alcohol | -0.765* (0.197) | -0.266 (0.160) | 0.391 (0.206) | -0.533* (0.129) | -0.246 (0.415) | 1.058+ (0.561) | -0.257 (0.150) | -0.001 (0.001) | -0.025 (0.022) | 0.007* (0.003) | 0.000 (0.002) | -0.392+ (0.219) | -0.004 (0.005) |
| Ever Smokes | 0.052 (0.175) | -0.003 (0.143) | -0.026 (0.183) | -0.102 (0.217) | -0.060 (0.375) | -0.104 (0.523) | 0.095 (0.238) | -0.000 (0.001) | 0.161*** (0.020) | -0.002 (0.003) | 0.002 (0.003) | 3.078*** (0.189) | 0.043*** (0.004) |
| **Polygenic Scores** | | | | | | | | | | | | | |
| Longevity PGS | -0.057 (0.195) | 0.059 (0.147) | -0.080 (0.199) | -0.064 (0.127) | -0.042 (0.230) | -0.455 (0.313) | -0.055 (0.156) | -0.001 (0.001) | 0.001 (0.012) | -0.002 (0.002) | 0.001 (0.002) | 0.127 (0.111) | 0.000 (0.002) |
| NCEB PGS | -0.238 (0.197) | -0.045 (0.160) | 0.483* (0.206) | -0.128 (0.129) | 0.151 (0.223) | -0.085 (0.318) | -0.063 (0.150) | -0.000 (0.001) | 0.018 (0.012) | -0.000 (0.002) | -0.001 (0.002) | 0.106 (0.124) | 0.002 (0.003) |
| AFB PGS | 0.052 (0.175) | -0.003 (0.143) | -0.026 (0.183) | 0.021 (0.118) | 0.111 (0.216) | -0.445 (0.302) | -0.109 (0.132) | -0.000 (0.001) | -0.007 (0.011) | -0.001 (0.002) | 0.000 (0.002) | -0.153 (0.108) | -0.000 (0.002) |
| **Socioeconomic Background** | | | | | | | | | | | | | |
| Years of Education | -0.025 (0.075) | -0.143** (0.067) | -0.081 (0.085) | -0.049 (0.055) | 0.138 (0.102) | 0.242+ (0.138) | 0.028 (0.060) | -0.000 (0.000) | -0.008 (0.005) | -0.001 (0.001) | 0.000 (0.001) | -0.220*** (0.049) | -0.003* (0.001) |
| Parental Years of Education | -0.082 (0.057) | -0.143** (0.047) | -0.106+ (0.061) | -0.049 (0.041) | 0.094 (0.071) | -0.140 (0.107) | -0.037 (0.047) | -0.001*** (0.000) | -0.010** (0.004) | -0.001 (0.001) | -0.002*** (0.000) | -0.085* (0.039) | -0.001 (0.001) |

*(Continued)*

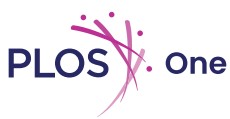

**Table 3.** (Continued)

| Variables | Model 1 Horvath 1 | Model 2 Hannum | Model 3 Levine | Model 4 Horvath 2 | Model 5 Lin | Model 6 Weidner | Model 7 Vidal-Bralo | Model 8 EpiTOC | Model 9 Zhang | Model 10 Bocklandt | Model 11 Garagnani | Model 12 GrimAge | Model 13 Dunedin-inPACE |
|---|---|---|---|---|---|---|---|---|---|---|---|---|---|
| 2014 Total of All Assets | −0.009 | −0.015 | −0.022* | −0.005 | −0.022 | −0.018 | −0.006 | −0.000 | −0.002** | 0.000 | 0.000 | −0.017* | −0.000 |
|  | (0.012) | (0.010) | (0.011) | (0.007) | (0.015) | (0.025) | (0.008) | (0.000) | (0.001) | (0.000) | (0.000) | (0.008) | (0.000) |
| 2014 Retirement Status (Ref. = Not retired) |  |  |  |  |  |  |  |  |  |  |  |  |  |
| Completely Retired | 0.411 | 0.051 | 0.604 | 0.312 | −0.193 | −0.315 | 0.237 | 0.001 | 0.032 | 0.005 | 0.003 | 0.616* | 0.000 |
|  | (0.405) | (0.319) | (0.464) | (0.288) | (0.487) | (0.681) | (0.299) | (0.001) | (0.029) | (0.005) | (0.004) | (0.273) | (0.006) |
| Partly Retired | 0.596 | 0.030 | 0.874+ | 0.348 | −0.325 | 0.350 | 0.314 | 0.001 | 0.011 | 0.002 | 0.002 | 0.380 | 0.001 |
|  | (0.457) | (0.379) | (0.520) | (0.329) | (0.572) | (0.797) | (0.360) | (0.001) | (0.032) | (0.005) | (0.004) | (0.328) | (0.007) |
| Question Irrelevant | −2.167* | −1.106 | −1.375 | −1.231+ | −2.035 | 0.878 | −0.140 | −0.003 | −0.003 | 0.004 | −0.015+ | 0.417 | 0.009 |
|  | (0.884) | (0.739) | (0.880) | (0.661) | (1.293) | (1.932) | (0.615) | (0.002) | (0.070) | (0.011) | (0.008) | (0.611) | (0.014) |
| Demographic Characteristics |  |  |  |  |  |  |  |  |  |  |  |  |  |
| Female | −1.055** | −2.055*** | −1.406*** | −1.142*** | −1.291*** | −1.167* | −1.953*** | 0.001 | −0.201*** | 0.021*** | 0.005+ | −3.244*** | −0.018*** |
|  | (0.326) | (0.267) | (0.369) | (0.225) | (0.391) | (0.555) | (0.244) | (0.001) | (0.021) | (0.004) | (0.003) | (0.203) | (0.004) |
| 2014 Age | 0.695*** | 0.758*** | 0.705*** | 0.776*** | 0.799*** | 0.552*** | 0.346*** | 0.000*** | 0.012*** | −0.003*** | 0.005*** | 0.658*** | −0.000 |
|  | (0.043) | (0.036) | (0.049) | (0.029) | (0.058) | (0.080) | (0.036) | (0.000) | (0.003) | (0.000) | (0.000) | (0.027) | (0.001) |
| Cohort (Ref. = Old) |  |  |  |  |  |  |  |  |  |  |  |  |  |
| Middle | 1.018+ | 1.028* | −0.264 | 1.171** | 0.519 | −1.263 | −0.332 | −0.001 | 0.000 | 0.003 | 0.011* | −0.465 | −0.006 |
|  | (0.571) | (0.476) | (0.612) | (0.381) | (0.704) | (0.942) | (0.448) | (0.002) | (0.036) | (0.006) | (0.005) | (0.361) | (0.008) |
| Young | −0.709 | 0.011 | −0.398 | 0.170 | −0.713 | 1.124 | −0.246 | −0.001 | −0.003 | 0.004 | 0.012 | −1.194* | −0.016 |
|  | (0.770) | (0.682) | (0.891) | (0.540) | (1.080) | (1.465) | (0.668) | (0.002) | (0.053) | (0.009) | (0.007) | (0.532) | (0.012) |
| 2014 Family Size | 0.308+ | 0.047 | 0.245 | −0.083 | 0.592** | 0.414 | 0.426** | 0.000 | 0.008 | −0.005* | −0.001 | 0.082 | 0.002 |
|  | (0.174) | (0.139) | (0.216) | (0.148) | (0.215) | (0.299) | (0.132) | (0.000) | (0.012) | (0.002) | (0.001) | (0.122) | (0.002) |
| 2014 Number of Living Siblings | 0.131 | −0.033 | −0.138 | −0.035 | 0.062 | 0.135 | 0.045 | 0.000 | 0.004 | −0.001 | −0.000 | −0.076 | −0.001 |
|  | (0.084) | (0.065) | (0.089) | (0.058) | (0.102) | (0.142) | (0.063) | (0.000) | (0.005) | (0.001) | (0.001) | (0.055) | (0.001) |
| Religious Affiliation (Ref. = Protestant) |  |  |  |  |  |  |  |  |  |  |  |  |  |
| Catholics | −0.164 | −0.640+ | −0.428 | 0.142 | 0.077 | −0.238 | −0.291 | −0.000 | −0.060* | −0.001 | 0.001 | 0.012 | −0.003 |
|  | (0.419) | (0.350) | (0.423) | (0.291) | (0.492) | (0.715) | (0.309) | (0.001) | (0.027) | (0.004) | (0.004) | (0.266) | (0.006) |
| None | 0.127 | −0.332 | 0.048 | −0.030 | −0.498 | 0.524 | 0.335 | 0.000 | 0.004 | −0.003 | −0.008* | 0.065 | −0.002 |
|  | (0.520) | (0.380) | (0.579) | (0.349) | (0.628) | (0.850) | (0.407) | (0.001) | (0.030) | (0.006) | (0.004) | (0.322) | (0.007) |

*(Continued)*

Table 3. (Continued)

| Variables | Model 1 Horvath 1 | Model 2 Hannum | Model 3 Levine | Model 4 Horvath 2 | Model 5 Lin | Model 6 Weidner | Model 7 Vidal-Bralo | Model 8 EpiTOC | Model 9 Zhang | Model 10 Bocklandt | Model 11 Garagnani | Model 12 GrimAge | Model 13 Dunedin-inPACE |
|---|---|---|---|---|---|---|---|---|---|---|---|---|---|
| Other | 0.420 | 1.040 | 0.771 | −0.535 | 0.660 | 1.498 | −0.244 | −0.001 | 0.024 | −0.007 | 0.005 | 0.976+ | −0.004 |
|  | (0.980) | (1.036) | (1.363) | (0.659) | (1.174) | (2.007) | (0.779) | (0.003) | (0.070) | (0.011) | (0.010) | (0.564) | (0.013) |
| Population Stratification |  |  |  |  |  |  |  |  |  |  |  |  |  |
| PC1 | 38.807* | 18.667 | 20.682 | 18.922 | 17.085 | −45.714 | −11.862 | 0.027 | 0.434 | 0.072 | 0.234 | −24.885* | −0.264 |
|  | (19.026) | (18.923) | (22.520) | (14.094) | (21.011) | (31.710) | (15.473) | (0.058) | (1.398) | (0.216) | (0.178) | (11.978) | (0.266) |
| PC2 | 11.323 | −9.943 | −7.477 | 9.235 | −17.424 | −20.886 | −3.243 | −0.064 | −1.449 | 0.216 | 0.126 | −1.919 | 0.145 |
|  | (16.494) | (12.887) | (17.568) | (11.592) | (19.922) | (30.003) | (13.226) | (0.050) | (1.053) | (0.177) | (0.130) | (10.187) | (0.227) |
| PC3 | 4.943 | −4.025 | −24.394 | −16.107 | −16.576 | 8.323 | −4.076 | −0.010 | 1.545 | 0.099 | 0.027 | 16.243 | 0.382+ |
|  | (17.321) | (14.021) | (18.322) | (11.894) | (22.833) | (28.363) | (13.088) | (0.046) | (1.156) | (0.187) | (0.146) | (10.286) | (0.225) |
| PC4 | −22.842 | −15.763 | −1.900 | −18.353 | 11.742 | 52.317+ | 0.569 | 0.007 | 0.802 | 0.165 | 0.136 | −0.133 | 0.202 |
|  | (17.096) | (13.351) | (18.627) | (11.628) | (20.992) | (30.706) | (13.971) | (0.041) | (1.177) | (0.194) | (0.144) | (10.958) | (0.249) |
| PC5 | −5.905 | 9.089 | −8.447 | 1.296 | −5.210 | −53.613 | −7.414 | −0.093 | −0.844 | −0.508+ | 0.208 | 13.038 | 0.195 |
|  | (24.455) | (22.517) | (27.358) | (17.187) | (28.778) | (40.939) | (20.573) | (0.067) | (1.632) | (0.266) | (0.222) | (13.952) | (0.315) |
| PC6 | −45.343* | −6.815 | −26.332 | −18.724+ | −42.671* | 21.097 | 9.605 | −0.019 | 0.787 | 0.377* | 0.013 | 3.361 | −0.083 |
|  | (18.353) | (13.909) | (17.423) | (11.260) | (20.019) | (27.096) | (13.153) | (0.047) | (1.048) | (0.179) | (0.149) | (10.765) | (0.237) |
| PC7 | −24.521 | −14.915 | −34.023+ | −2.992 | −28.179 | 1.423 | −7.828 | 0.004 | 0.261 | −0.026 | −0.009 | 4.711 | 0.187 |
|  | (17.449) | (13.458) | (19.285) | (11.495) | (20.480) | (28.641) | (14.742) | (0.044) | (1.099) | (0.190) | (0.151) | (10.946) | (0.237) |
| PC8 | −23.338 | −11.405 | −1.748 | 0.893 | −26.476 | 11.356 | −18.138 | −0.080+ | −0.547 | 0.047 | 0.117 | 14.113 | 0.255 |
|  | (16.649) | (12.933) | (17.862) | (11.945) | (19.484) | (29.110) | (12.619) | (0.045) | (1.167) | (0.187) | (0.156) | (11.122) | (0.244) |
| PC9 | 13.759 | 25.581+ | 3.865 | 17.885+ | 21.909 | −39.141 | 13.026 | 0.062 | 1.793+ | −0.181 | 0.036 | 4.263 | 0.036 |
|  | (16.419) | (13.532) | (17.173) | (10.850) | (19.757) | (28.066) | (13.218) | (0.045) | (1.088) | (0.170) | (0.137) | (10.051) | (0.218) |
| PC10 | −3.749 | −9.254 | −2.725 | 3.657 | −10.995 | −19.454 | −10.050 | 0.058 | −0.550 | 0.062 | −0.081 | −10.823 | −0.190 |
|  | (19.831) | (14.763) | (18.479) | (12.671) | (21.234) | (30.186) | (14.671) | (0.050) | (1.118) | (0.191) | (0.161) | (10.239) | (0.230) |
| Constant | 20.786*** | 6.169* | 12.378*** | 19.923*** | 1.877 | 26.887*** | 41.055*** | 0.047*** | −1.632*** | 0.615*** | 0.358*** | 28.076*** | 1.137*** |
|  | (3.656) | (2.954) | (3.967) | (2.440) | (5.024) | (6.629) | (2.939) | (0.010) | (0.232) | (0.039) | (0.031) | (2.313) | (0.050) |
| Observations | 1,654 | 1,654 | 1,654 | 1,654 | 1,654 | 1,654 | 1,654 | 1,654 | 1,654 | 1,654 | 1,654 | 1,654 | 1,654 |
| Adjusted R-squared | 0.537 | 0.663 | 0.515 | 0.741 | 0.479 | 0.162 | 0.327 | 0.0723 | 0.187 | 0.164 | 0.445 | 0.775 | 0.0866 |

Standard errors (in parentheses) are bias-corrected and accelerated (BCa) bootstrap standard errors based on 1,000 replications.

*** p<0.001, ** p<0.01, * p<0.05, + p<0.1.

CESD scores across models, suggesting that experiencing a marital transition may be linked to poorer mental health in later life (e.g., Hannum model: b = 0.614, SE = 0.191, p < .01).

Gender-stratified analyses are shown in S5 Table. Among men, marital status change was consistently associated with about a 0.81-point increase in CESD across clock models and remained statistically significant (e.g., Hannum model: b = 0.811, SE = 0.310, p < .01). Among women, the marital status change coefficients were smaller (around 0.41) and not statistically significant (e.g., Hannum model: b = 0.406, SE = 0.261, p > .05). The pooled models that interact gender with all covariates also confirm that the marital status change coefficient differs significantly by gender. This sex contrast indicates that the average mental health impact of marital change is concentrated among men. Furthermore, although the full models did not show significant main effects of marital status in 2016 on CESD scores in 2020, marital status in 2016 was associated with CESD among men in the gender-stratified models. Specifically, men who were separated or divorced in 2016 reported significantly higher CESD scores in 2020 compared to married or partnered men, and widowhood was also linked to elevated symptoms (Hannum model (males, main-effects model): Separated/Divorced: b = 0.845, SE = 0.358, p < .05; Widowed: b = 0.812, SE = 0.392, p < .05). These marital-status differences were not statistically significant among women, which aligns with previous findings suggesting that marital dissolution may have more severe mental health consequences for men than for women.

To test Hypothesis 2, which posits that epigenetic clocks moderate the effect of marital status change on mental health, interaction terms between each clock and marital status change were added in models shown in S4 and S5 Tables. In the full sample (S4 Table), none of the clock × marital change interaction terms reached statistical significance at p < 0.05. In sex-stratified interaction models (S5 Table), the male models showed a statistically significant interaction for the Garagnani clock (Garagnani × marital change among men: b = −10.638, SE = 4.249, p < .05), whereas the interaction terms are not statistically significant in the female models. Although the full-sample interaction terms were not statistically significant, the male-stratified results provide limited but suggestive evidence consistent with a buffering pattern, with statistically significant moderation for Garagnani and no corresponding evidence among women.

Fig 1 graphically illustrates these findings for the male sample. The difference in CESD scores between those who experienced marital status change (solid line) and those who did not (dashed line) was larger among biologically younger men compared to biologically older men. The gap narrowed and reversed when epigenetic age on the clock scale reached approximately 80. Below this epigenetic age, men who experienced marital status change reported worse mental health than those who did not, whereas above this epigenetic age, little difference or even slightly better mental health was observed among those who experienced a change. These findings suggest that marital status change is more detrimental to biologically younger men, supporting the moderating role of epigenetic aging.

As a robustness check, we separated marital transitions into union dissolution (including widowhood) and union formation using a three-category transition measure. In these models, the transition pattern is concentrated in union dissolution, while union formation estimates are unstable due to the small number of observed union formations in this subsample. In the male CESD interaction models (S1 File Supplementary Excel Table SE9), union dissolution coefficients are generally positive across clocks, and the clock × union dissolution interactions mostly follow the same moderation pattern as in the main models, whereas the union-formation main effects and interaction terms are imprecisely estimated.

### Epigenetic aging, marital status change, and mortality risk

Cox proportional hazards models estimating mortality risk are reported in S6 through S8 Tables. As expected, higher epigenetic ages were associated with increased mortality risk after adjusting for covariates. Significant positive associations were observed for the Hannum, Levine, Vidal-Bralo, Zhang, GrimAge, and DunedinPACE clocks (Hannum: b = 0.022, SE = 0.010, p < .05; Levine: b = 0.035, SE = 0.007, p < .001; Vidal-Bralo: b = 0.026, SE = 0.009, p < .01; Zhang: b = 0.872, SE = 0.125, p < .001; GrimAge: b = 0.109, SE = 0.012, p < .001; DunedinPACE: b = 3.063, SE = 0.632, p < .001). For instance, a one-year increase in the Levine clock was associated with a 3.6% increase in the hazard of death [exp(0.035) = 1.036].

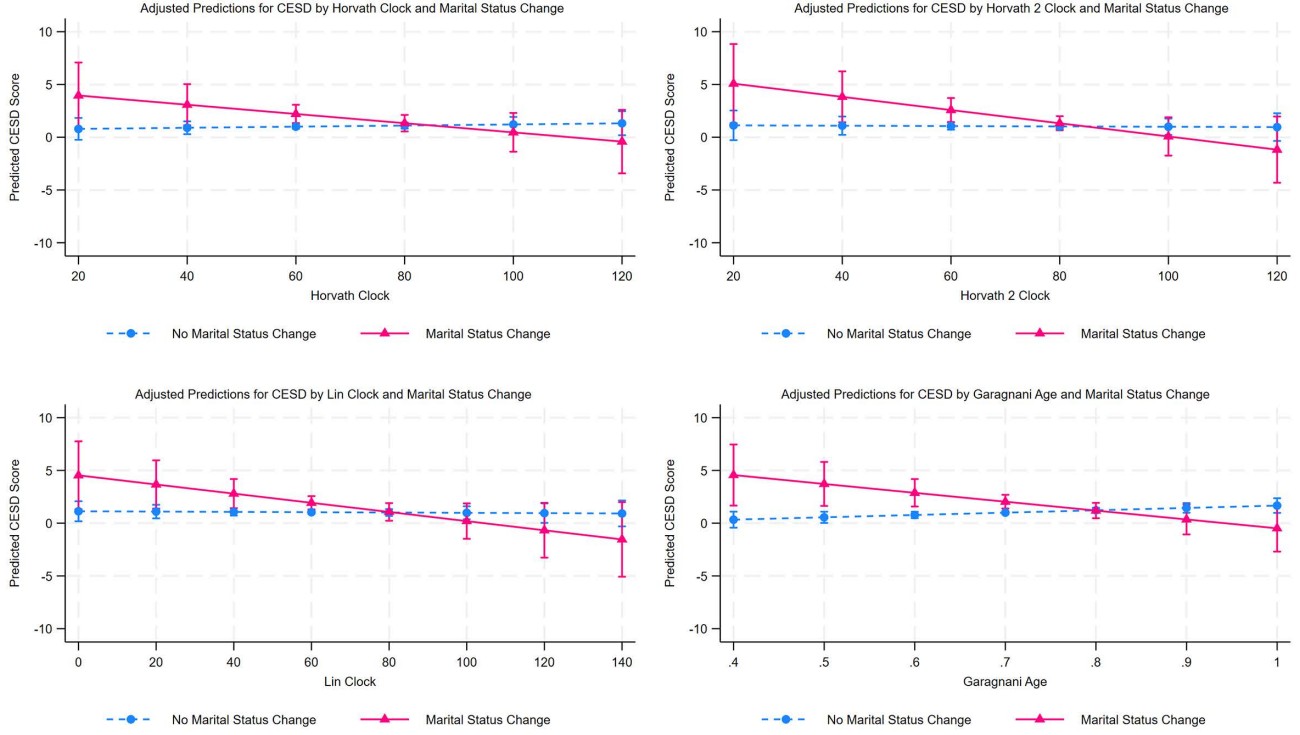

**Fig 1. Adjusted Predictions for CESD Score in 2020 by Epigenetic Clocks (2016) and Marital Status Change Between 2016 and 2020 (Males).**
Adjusted predictions of CESD score in 2020 by epigenetic clocks (2016) and marital status change between 2016 and 2020 among men. Solid lines indicate marital status change; dashed lines indicate no change. In the male interaction models (S5 Table), the Garagnani × marital change interaction is statistically significant at p < .05, while the corresponding interactions for the other clocks are marginally significant at p < .10. Predictions are based on the fully adjusted models reported in S5 Table.

Similar patterns were found in both male and female models reported in S8 Table, with GrimAge, Zhang, and Dunedin-PACE significant predictors of mortality for both men and women (males: Zhang: b = 0.976, SE = 0.222, p < .001; Grim-Age: b = 0.112, SE = 0.018, p < .001; DunedinPACE: b = 4.061, SE = 0.938, p < .001; females: Zhang: b = 0.840, SE = 0.169, p < .001; GrimAge: b = 0.112, SE = 0.019, p < .001; DunedinPACE: b = 2.110, SE = 0.902, p < .05).

In contrast to the strong associations observed for epigenetic clocks, marital status change was not significantly associated with mortality risk in the full models, although the coefficients were positive (e.g., Levine: b = 0.249, SE = 0.170, p = .14; S6 Table). Thus, while marital transitions may impact mental health, their direct relationship with mortality appeared less pronounced once biological aging was considered.

S7 Table examines the moderating effects of epigenetic aging on the relationship between marital status change and mortality risk to test Hypothesis 2. Significant interaction effects were found for the Hannum, Garagnani, and Grim-Age clocks in the full sample (S7 Table), with negative interaction terms suggesting that accelerated biological aging may attenuate the association between marital status change and mortality risk (Hannum × marital change: b = −0.038, SE = 0.016, p < .05; Garagnani × marital change: b = −4.536, SE = 1.991, p < .05; GrimAge × marital change: b = −0.044, SE = 0.020, p < .05). In sex-stratified models (S8 Table), significant moderating patterns were concentrated among men, with statistically significant interactions for Hannum, Zhang, and GrimAge (Hannum × marital change: b = −0.065, SE = 0.028, p < .05; Zhang × marital change: b = −1.520, SE = 0.521, p < .01; GrimAge × marital change: b = −0.079, SE = 0.032, p < .05). Even in models where the interaction terms were not statistically significant, the interaction coefficients in the male models were generally in the hypothesized negative direction. Among women, no consistent moderating

effects were found, except for an interaction using the Zhang clock, which was in the opposite direction (Zhang × marital change: b = 1.090, SE = 0.502, p < .05). Gender-interaction tests confirm that some marital-status coefficients and the Zhang-based moderation differ significantly between men and women, consistent with the stronger and more coherent moderation pattern observed among men.

Fig 2 graphically presents the interaction pattern for the male sample. Survival functions were plotted for men with biological ages of 65 and 85 years, comparing those who experienced marital status change to those who did not. For Zhang, survival functions were plotted at the 75th and 95th percentiles (P75 and P95) because it is not on an age scale. At younger epigenetic ages (65), men who experienced a marital transition had lower survival probabilities than those who did not. However, at older epigenetic ages (85), men who experienced marital status change had similar or even better survival probabilities compared to those without a marital transition. This pattern is consistent with the negative clock × marital change interactions observed in the male models, particularly for Hannum, Zhang, and GrimAge. Overall, these results provide evidence consistent with Hypothesis 2, suggesting a moderating role of epigenetic aging in the association between marital status change and mortality risk. The findings also align with Hypothesis 3, suggesting that marital status transitions may be more strongly associated with mortality risk among men than among women, and that accumulated life experiences may contribute to greater resilience among men.

We also estimated a three-category transition specification distinguishing no change, union dissolution (including widowhood), and union formation as robustness checks. Similar to CESD, the union-formation effects are imprecisely

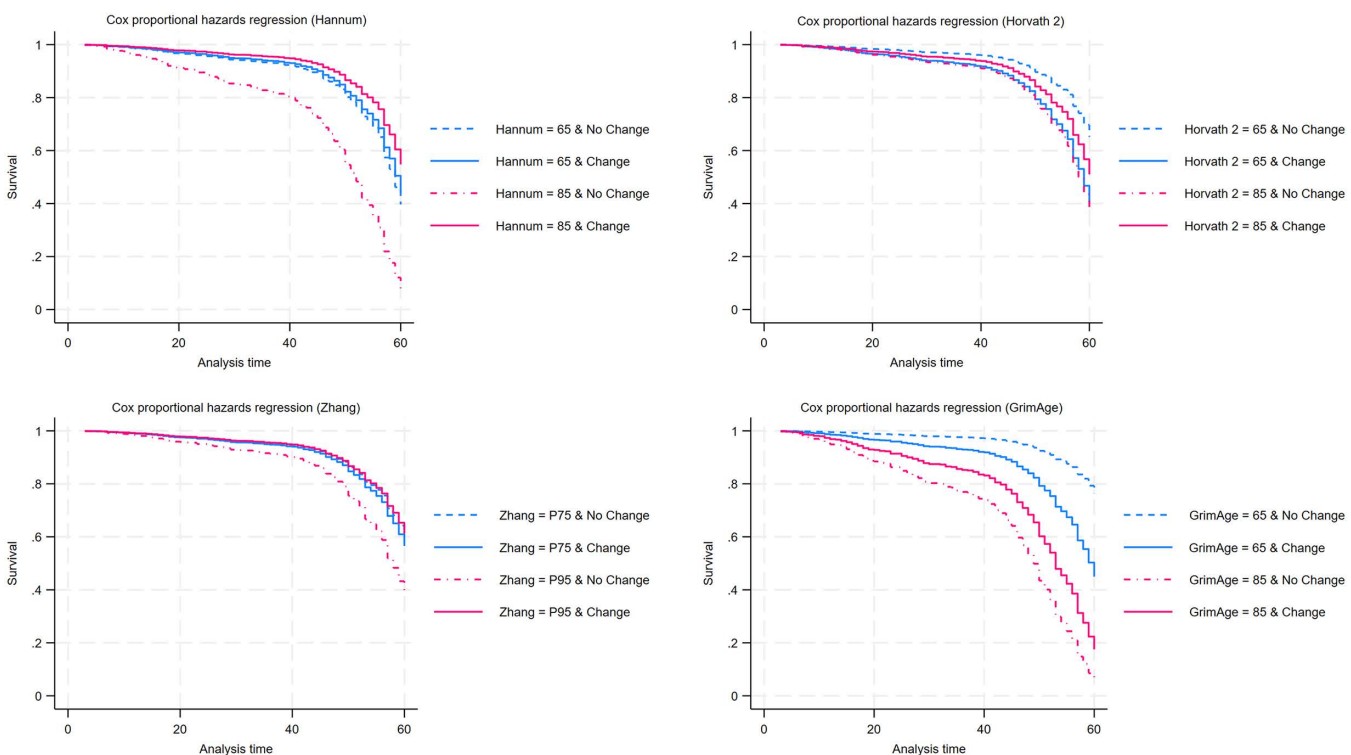

**Fig 2. Adjusted Survival Functions Through Follow-Up at the 2020 Interview from Cox Models by Epigenetic Clocks and Marital Status Change (Males).** Solid lines indicate marital status change; dashed lines indicate no change. Curves are plotted at epigenetic ages 65 and 85 for Hannum, Horvath 2, and GrimAge, and at the 75th and 95th percentiles (P75 and P95) for Zhang because it is not on an age scale. In the male Cox interaction models (S8 Table), the clock × marital change interactions are significant for Hannum, Zhang, and GrimAge (p < .05) and marginally significant for Horvath 2 (p < .10), consistent with a buffering pattern at higher levels of epigenetic aging.



estimated in this subsample, while the clock × union dissolution interactions generally mirror the main moderation pattern. In the male Cox interaction models (S1 File Supplementary Excel Table SE16), the union-formation main effects and interaction terms remain unstable, whereas the clock × union dissolution interactions remain negative for the clocks driving the main findings (union dissolution × Zhang: b = −1.706, SE = 0.597, p < .01; union dissolution × GrimAge: b = −0.079, SE = 0.039, p < .05).

## Robustness checks

To assess the robustness of our findings, we conducted several additional analyses (see Supplementary Tables S1-S8 Tables and S1 File Supplementary Excel Tables). First, we re-estimated the CESD and mortality models using a three-category transition measure (no change, union dissolution including widowhood, and union formation). The transition signal was concentrated in union dissolution, while union-formation estimates were unstable due to sparse events (20 cases). In the male CESD interaction models (S1 File SE9) and male Cox interaction models (S1 File SE16), the clock × union dissolution interactions largely mirrored the primary moderation pattern, whereas union-formation main effects and interaction terms were imprecisely estimated.

Second, because thirteen epigenetic clocks were tested across multiple outcomes, we applied Bonferroni and false discovery rate (FDR) corrections to account for multiple comparisons. Both the main effects and interaction terms were re-evaluated using adjusted significance thresholds. As shown in the supplementary tables, the only interaction term that remained statistically significant after Bonferroni correction was the Zhang clock × marital-status-change interaction in the male mortality models, and this interaction also remained significant under FDR correction, consistent with the primary moderation pattern.

Third, we addressed item-specific missingness and potential selection using multiple imputation and inverse probability weighting. We estimated multiple imputation models for the epigenetic aging analyses, the CESD main and interaction models, and the Cox models, with gender-stratified versions reported alongside the pooled results in the S1 File Supplementary Excel Tables. Estimates were generally consistent with the complete-case results in direction and interpretation, although some interaction terms were less precisely estimated. We also implemented inverse probability weighting in two ways. The first set of weights adjusts for being observed at the baseline wave by modeling the probability of baseline observation using earlier-wave demographic, socioeconomic, and health measures. The second set of weights adjusts for being observed in the DNA methylation and genotype subsamples among respondents otherwise eligible for analysis. We then re-estimated the primary models using these weights, again with gender-stratified specifications reported in the S1 File Supplementary Excel Tables. Across these IPW checks, the main effects and the core moderation pattern remained similar to the primary findings, though precision was reduced in several interaction models, which is expected under weighting.

Fourth, child-related measures and child-based social support can be substantively important in later life because children often represent a primary source of social support and may be linked to mental health and mortality risk. At the same time, these measures are closely intertwined with marital status and later-life family processes, raising the risk of overcontrol when included alongside marital status in observational models. Recent discussions of control-variable selection emphasize the need to be explicit about adjustment choices and to avoid conditioning on variables that may be highly confounded with, or downstream of, focal predictors [63]. For this reason, we report the parsimonious specifications in the main text and present child-adjusted models as sensitivity analyses in the S1 File Supplementary Excel Tables. Full results are reported in the S1 File Supplementary Excel Tables, including the bootstrapped epigenetic aging models, the CESD main and interaction models, and the Cox main and interaction models that include these child-related covariates. The substantive conclusions were similar to the primary models, and the main moderation pattern was preserved. Because child-based measures are a major source of item-specific missingness, models that include them can further restrict the analytic sample, potentially shifting it toward respondents with more complete kin information and stronger family ties.

## Discussion and conclusions

Chronological age is often treated as a neutral marker, yet it carries social meanings and reflects lived experience that can become biologically embedded over the life course [1,14,64,65]. This study suggests that the socially constructed process of aging is embedded within biological systems. By examining the social underpinnings of biological aging and considering epigenetic clocks as markers of accumulated life experiences, four primary findings emerge. First, marital status showed limited and clock-specific associations with epigenetic aging in models that adjusted for genetic factors and other social covariates. Second, marital status change within the previous four years was associated with higher CESD scores but was not significantly associated with mortality risk. Among the thirteen epigenetic clocks, CESD score was significantly associated with Hannum and GrimAge, whereas mortality risk was significantly predicted by Hannum, Levine, Vidal-Bralo, Zhang, GrimAge, and DunedinPACE. Third, evidence that epigenetic clocks moderated responses to marital status change was selective and outcome-specific. For CESD score, no clock × marital-change interactions were statistically significant in the full sample. For mortality, buffering interactions were most evident for clocks including Hannum, Garagnani, and GrimAge. Fourth, these moderation patterns were more evident among men, suggesting potential gender differences in how marital transitions intersect with biological aging and later-life vulnerability.

Consistent with previous research [12,13,20,21,37,38,66], our study supports the view that epigenetic aging is malleable. Even after controlling for chronological age and genetic predispositions, social factors such as marital status and health behaviors were associated with biological aging. However, these associations were not uniform across clocks, and marital-status differences were most apparent for a limited set of clock measures. In sex-stratified models, several marital-status contrasts also differed by sex: among women, being separated or divorced was associated with a lower Weidner estimate and widowhood was associated with higher DunedinPACE, whereas among men, widowhood was associated with a lower Bocklandt estimate (see S2 Table). Because the Bocklandt clock is often oriented differently from other clocks [20], this association may not map straightforwardly onto "younger" versus "older" aging in the same way as other clocks. Although significant associations were observed for a handful of the epigenetic clocks analyzed, this heterogeneity is informative rather than unexpected because clocks differ in their training targets and thus capture partly distinct dimensions of biological aging and health risk [18,48]. Relatedly, epigenetic clocks are derived from DNA methylation profiles, which are responsive to social and behavioral contexts, and the observed associations suggest that some clock constructs are more sensitive to socially patterned exposures and behaviors than others in this setting [12,21,22]. In this sense, epigenetic clocks may serve as integrated markers of biological, social, and psychological aging, capturing cumulative life experiences that could include adaptive responses to adversity [12,48].

The clock-specific patterns in this study are consistent with differences in what each measure was designed to capture and the pathways most closely tied to the outcome. In the CESD models, statistically significant associations were concentrated in Hannum and GrimAge, whereas mortality risk was predicted by a broader set of measures including GrimAge and DunedinPACE. Several clocks used here produce an age-like estimate and are often interpreted in terms of age acceleration, meaning how much older or younger the methylation profile appears relative to chronological age. In contrast, DunedinPACE is designed to capture the pace of aging, meaning whether aging-related physiological change is occurring faster or slower, rather than providing an age-like level of biological aging [67]. GrimAge was developed to predict mortality risk and incorporates methylation-based surrogates for smoking pack-years and multiple plasma proteins, which index exposure and physiological risk processes that are also implicated in depressive symptoms [68]. Consistent with this interpretation, prior work in older adults reports that GrimAge acceleration is associated with depressive symptoms, including evidence from analyses based on HRS [69]. Evidence also links epigenetic age acceleration measures to depressive symptoms in other cohort studies [70]. These distinctions help explain why some epigenetic measures align more closely with symptom variation, while others map more clearly onto survival risk.

Beyond using epigenetic clocks as biomarkers of physiological deterioration, this study proposes that they also function as indicators of life experience accumulation. Clocks that more consistently predicted mortality risk also showed clearer

evidence of buffering vulnerability to recent marital status change in the mortality models. For CESD score, moderation effects were more selective, with significant evidence in the male stratified models for the Garagnani clock. This finding resonates with the adage "what does not kill us makes us stronger," suggesting that accumulated experiences may enhance resilience in later life. These results suggest that while aging brings vulnerabilities, it also brings strengths, including accumulated knowledge, coping skills, and emotional regulation. Viewing aging as a resource accumulation process offers a novel perspective on resilience in later life.

This interpretation is supported by theoretical frameworks on resilience, which emphasize the interplay of psychological, social, and biological factors in adaptive functioning [15–17]. Resilience develops not only from personal characteristics such as temperament or emotional stability but also from life experiences that foster coping strategies and emotional regulation. In addition to these psychological resources, researchers increasingly recognize a biological dimension of resilience, seen in the body's capacity to adapt to stress at the physiological level. Findings from recent research [44–46] indicate that resilience-related traits such as self-control and emotion regulation can buffer the negative impact of cumulative stress on epigenetic aging. Related reviews further link trauma, discrimination, and chronic adversity to accelerated epigenetic aging, while also emphasizing that psychosocial resources shape biological embedding across the life course [45,48,49]. At the same time, resilience processes may not manifest uniformly across all epigenetic clock constructs or health outcomes, given differences in what clocks capture and how outcomes reflect downstream risk [18,47,48]. In this light, even if evidence of moderation is not uniform across outcomes or clocks, the epigenetic moderation results in this study are consistent with the idea that resilience can be biologically embedded. Accumulated life experiences may strengthen the body's capacity to adapt to and withstand future adversities.

Importantly, our approach differs from prior research on resilience and epigenetic aging by treating epigenetic clocks as potential effect modifiers rather than as outcomes. Whereas psychological resilience has been modeled as a predictor of epigenetic aging in Zhang et al.'s HRS analysis [46], and resilience-related traits have been shown to moderate stress-related epigenetic aging [44], we test whether epigenetic aging profiles condition responses to a later-life transition, recent marital status change, using CESD score and mortality as outcomes. These approaches address different parts of the same resilience process. In Zhang et al. [46], higher psychological resilience was associated with slower epigenetic aging, which can be interpreted as lower biological vulnerability. In our analyses, epigenetic aging profiles are used to stratify vulnerability and adaptation, and we find that moderation is more apparent for mortality than for CESD score. In particular, Zhang et al. found that resilience was most robustly associated with slower epigenetic aging for clocks including Levine, Zhang, Garagnani, and GrimAge [46], whereas in our analyses moderation was clearest in the mortality models and concentrated in a smaller subset of clocks, with Hannum, Garagnani, and GrimAge showing the strongest evidence. This outcome contrast is plausible given that several clocks in common use were developed to capture healthspan or mortality risk (e.g., GrimAge and Levine/PhenoAge) or the pace of aging (DunedinPACE), and these measures are often more tightly linked to survival outcomes than clocks trained primarily on chronological age [18,20,47–49].

Analyses using gender-stratified samples revealed important gender differences in responses to marital status change and in the moderating effects of epigenetic clocks. Overall, marital status change and its interactions with epigenetic clocks were more consistently evident in men's models than in women's models. These findings are consistent with prior research showing that men may suffer more immediate emotional consequences following marital dissolution [54,71], and with gendered accounts of marriage as a source of social regulation and support that may be especially consequential for men [33,35]. Specifically, marital status change was associated with higher CESD scores more clearly among men than women. In addition, the clearest evidence of moderation appeared in men's mortality models, most notably for Hannum and GrimAge, with Zhang also showing moderation among men.

However, the observation that older biological age buffered the impact of marital transitions for men suggests that, over time, men appear to recover more fully, particularly when accumulated life experiences provide critical resilience resources. This finding aligns with results from Leopold T. [72], who showed that although men experienced greater

immediate declines in well-being following divorce, they tended to recover faster than women. In contrast, women experienced prolonged financial setbacks that were slower to resolve. Thus, while men may be more vulnerable in the short term, they appear to rebound more fully when sufficient psychological, social, and biological resources are available.

As a final note, this study is observational in nature. While the analyses identify statistical associations, the results should not be interpreted as evidence of causal relationships. In addition, the moderation patterns varied across outcomes and clocks, and future work with finer measurement around transitions will help clarify when and for whom resilience processes are most evident.

Although this study offers important insights, it is not without limitations. First, we could not implement a fully detailed relationship-trajectory approach in this study, such as separating remarriage from continuous marriage or modeling finer partnership sequences. Trajectory-based typologies can be informative for understanding later-life relationship histories [73]. However, in our analytic sample with DNA methylation and polygenic scores, union formation and remarriage events are rare within the study windows. Reconstructing remarriage timing from the longitudinal marital-history files would also yield very small cells once clock interactions and sex stratification are incorporated. Future work with larger epigenetic samples and longer observation windows will be better positioned to test whether clock moderation differs for remarriage versus continuous marriage and for union formation versus dissolution.

Second, our marital transition measure captures status changes observed across the HRS waves used to define the study windows (2016–2018–2020 for depressive symptoms and 2012–2014–2016 for mortality). This approach does not allow us to distinguish dissolutions that occurred long before the observation period from more recent later-life divorces or bereavement, nor does it model time since divorce or widowhood. Meta-analytic evidence suggests that the health consequences of dissolution vary by recency, with mortality risks often most elevated shortly after spousal loss and changing over longer follow-up periods [41–43]. Future studies with richer marital-history timing and larger epigenetic samples will be better positioned to test whether clock moderation differs for recent versus long-ago dissolutions.

Third, the precise nature of what epigenetic clocks measure remains uncertain. While they can be conceptualized as summaries of cumulative life experiences, the threshold at which accumulated experience becomes detrimental rather than protective remains unclear. Furthermore, given that clocks are based on DNA methylation derived from blood samples, it is unclear how well they reflect aging processes in other tissues or biological systems. Greater understanding of the underlying mechanisms of epigenetic aging is needed.

Fourth, although we controlled for polygenic scores related to longevity and mental health, it did not control directly for polygenic scores specific to epigenetic clocks themselves, which were unavailable. Future research incorporating these genetic components will be necessary to better isolate the contributions of environmental and social factors on biological aging.

Fifth, the analytic sample was restricted to individuals of European ancestry, given that the epigenetic clocks and polygenic scores used were primarily developed in European-descent populations. The exclusion of minority groups limits generalizability, and future studies should strive to include more diverse populations, particularly considering the intersectionality of race, ethnicity, and societal inequalities.

Sixth, the analysis remains conditional on being observed in the DNA methylation and genotype components and on having nonmissing measures for the focal constructs. These requirements can lead to a positively selected analytic sample, potentially underrepresenting respondents who are in poorer health or less socially connected. Item-specific missingness and attrition may also be systematic, which can affect both generalizability and interpretation. Although we conducted multiple imputation and inverse probability weighting robustness checks and examined alternative adjustment sets in supplementary analyses, these approaches rely on additional assumptions and do not fully eliminate concerns about selective participation and attrition. The results should therefore be interpreted as conditional associations within the observed biomarker and genotype subsamples.

Finally, although we sought to integrate biological and social perspectives on aging, further theoretical development is needed to fully conceptualize biological age within a socially constructed framework. Future work should explore how biological and social aging processes interact across the life course, particularly how resilience is shaped by cumulative exposures and resources.

Despite these limitations, this study advances understanding by integrating epigenetic, genetic, and social measures. It demonstrates that epigenetic clocks can be conceptualized as markers of accumulated life experiences and highlights how these experiences may function as resilience resources in later life. Recognizing aging as both a biological and social process, and as a potential source of strength rather than inevitable decline, represents an important step forward in aging research.

## Supporting information

**S1 Table. Ordinary Least Squares Models Using Marital Status, PGSs, and Social Factors to Predict Epigenetic Clocks (HRS).**
(PDF)

**S2 Table. Ordinary Least Squares Models Using Marital Status, PGSs, and Social Factors to Predict Epigenetic Clocks for Males and Females (HRS).**
(PDF)

**S3 Table. Ordinary Least Squares Models Using Epigenetic Clocks and Marital Status Change to Predict CESD Score in 2020 (HRS).**
(PDF)

**S4 Table. Ordinary Least Squares Models Using Interaction Term Between Epigenetic Clocks and Marital Status Change to Predict CESD Score in 2020 (HRS).**
(PDF)

**S5 Table. Ordinary Least Squares Models Using Interaction Term Between Epigenetic Clocks and Marital Status Change to Predict CESD Score in 2020 for Males and Females (HRS).**
(PDF)

**S6 Table. Cox Proportional Hazards Regression Models Using Epigenetic Clocks and Marital Status Change to Predict Mortality Risk (HRS).**
(PDF)

**S7 Table. Cox Proportional Hazards Regression Models Using Interaction Term Between Epigenetic Clocks and Marital Status Change to Predict Mortality Risk (HRS).**
(PDF)

**S8 Table. Cox Proportional Hazards Regression Models Using Interaction Term Between Epigenetic Clocks and Marital Status Change to Predict Mortality Risk for Males and Females (HRS).**
(PDF)

**S1 File. Supplementary Excel Tables.**
(XLSX)

## Author contributions

**Conceptualization:** Meng-Jung Lin.

**Data curation:** Meng-Jung Lin.



**Formal analysis:** Meng-Jung Lin.

**Funding acquisition:** Meng-Jung Lin.

**Investigation:** Meng-Jung Lin.

**Methodology:** Meng-Jung Lin.

**Project administration:** Meng-Jung Lin.

**Resources:** Meng-Jung Lin.

**Software:** Meng-Jung Lin.

**Supervision:** Meng-Jung Lin.

**Validation:** Meng-Jung Lin.

**Visualization:** Meng-Jung Lin.

**Writing – original draft:** Meng-Jung Lin.

**Writing – review & editing:** Meng-Jung Lin.

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
