## [Decision Letter · Decision Letter 0]

23 Dec 2025

Dear Dr. Lin,

Thank you for submitting your manuscript to PLOS ONE. After careful consideration, we feel that it has merit but does not fully meet PLOS ONE’s publication criteria as it currently stands. Therefore, we invite you to submit a revised version of the manuscript that addresses the points raised during the review process.

We look forward to receiving your revised manuscript.

Kind regards,

Xiangwei Li

Academic Editor

PLOS One

**Journal Requirements:**

“This research was supported by a grant from the National Science and Technology Council of Taiwan, grant number 113-2628-H-002-016-.”

5. Please include captions for your Supporting Information files at the end of your manuscript, and update any in-text citations to match accordingly. Please see our Supporting Information guidelines for more information: http://journals.plos.org/plosone/s/supporting-information....

Reviewers' comments:

Reviewer's Responses to Questions

**Comments to the Author**

1. Is the manuscript technically sound, and do the data support the conclusions?

Reviewer #1: Partly

Reviewer #2: Partly

2. Has the statistical analysis been performed appropriately and rigorously?

Reviewer #1: Yes

Reviewer #2: Yes

3. Have the authors made all data underlying the findings in their manuscript fully available?

Reviewer #1: Yes

Reviewer #2: Yes

4. Is the manuscript presented in an intelligible fashion and written in standard English?

Reviewer #1: Yes

Reviewer #2: Yes

Reviewer #1: Using data from 1,449 non-Hispanic White participants in the Health and Retirement Study, the study analyzed thirteen epigenetic clocks to assess associations between marital transitions, depressive symptoms, and mortality, adjusting for genetic and social factors. Interaction terms tested whether epigenetic clocks moderated these associations. The findings support the notion that epigenetic clocks capture effects of accumulated life experiences, which include those that reflect deterioration and resilience. The author proposes that the current study expands the literature on epigenetic clocks to include psychosocial adaptation or resilience.

The topic is of interest, the paper is generally well written and the analyses seem appropriate. My main concerns are that the paper would benefit from a more rigorous connection to the existing literature as some sentences do not have supporting references to the literature and, even more saliently, the idea of psychological resilience in epigenetic clocks has been documented in the literature (see a couple cites below). Moreover, it would be appropriate to discuss the findings in light of prior research on the association between marriage and health, which shows marriage has health benefits for men. Second, it would be useful to compare and contrast this study’s findings to those of Zhang, who also examined epigenetic clocks in HRS.

https://pmc.ncbi.nlm.nih.gov/articles/PMC10828333/

https://www.nature.com/articles/s41398-021-01735-7

Specific Text comments:

#1 PAGE 3 Check grammatical structure of this sentence.

For instance, using data from the

52 National Longitudinal Study of Adolescent to Adult Health (Add Health), (16) showed that

53 educational attainment, income, health behaviors, and stress exposure robustly predict variation

54 in epigenetic age acceleration across multiple clocks.

#2 PAGE 4

“Although aging is often portrayed as a period of vulnerability,”

This does not read quite right. Given aging is an action, it’s not a life stage. Perhaps the author means that older age is a period of vulnerability or that aging heightens biological vulnerability to dysfunction.

#3 RESULTS SECTION

The results section needs to report estimates in text, using a reporting style such as AMA or APA.

#4 PAGE 25

“ Age may not be as neutral a concept as it initially appears.”

Please revise or better support the comment. Aging is often considered a negative effect and associated with negative stereotypes in US contexts.

#5 PAGE 25

The two passages pasted below raise two issues. First, the conceptualization of the clocks is positioned as both biological (second passage) and ‘more than biological’ (first passage). The position ought to be consistent throughout the paper and whichever position is used it ought to be well reasoned and supported by the extant literature. Second, the author presents a useful description of the clocks and their constructure in the methods but in the first passage appears to dismiss that there are differences in the clocks and that half showed significant associations and half did not. These points ought not to be dismissed but rather discussed. Moreover, specifically how the findings support the assertion that the clocks reflect social and psychological effects needs to be better articulated.

Although significant associations were observed for fewer than half of the

459 epigenetic clocks analyzed, the findings suggest that epigenetic aging reflects more than simple

460 biological deterioration. Rather, epigenetic clocks may serve as integrated markers of biological,

461 social, and psychological aging, capturing cumulative life experiences that could include

462 adaptive responses to adversity.

VERSUS

PAGE 28

In this light, the epigenetic moderation effects found in this study further

482 validate the idea that resilience is biologically embedded, with accumulated life experiences

483 strengthening the body's capacity to withstand future adversities.

#7

Higher resolution images would be useful.

Reviewer #2: This methodologically sophisticated and well-written manuscript uses HRS data to examine the extent to which multiple epigenetic clock measures moderate the association between marital statuses/transitions and health outcomes: depressive symptoms and mortality. The study draws on resilience theories, and the results are presented clearly and concisely. Despite these strengths, I have a number of concerns regarding the paper’s conceptual framing and method.

1. The concept of resilience is highly debated and critiqued. First, there are multiple perspectives on its meaning. Is it the capacity to withstand adversity with NO negative health effects? Or, is resilience the experience of health symptoms following a stressful event but then a bouncing back shortly thereafter? I encourage you to say more about how you are hypothesizing resilience effects. Second, the very concept is debated because of its emphasis on individual-level strengths without comparable attention to the structural factors that enable one to ‘bounce back’ or ‘withstand’. These structures include economic resources, white privilege, socially/economically advantaged network members who have the wherewithal to provide support, and more. These are important concerns, especially when studying marriage. Marriage is also a highly stratified institution. Persons with better health and economic resources are more likely to ever marry, and to remain married. Financial problems destabilize marriages and increase the risk of divorce. Likewise, low SES is a risk factor for premature mortality, and thus widowhood – given that spouses share a social location. Cumulative dis/advantage or weathering frameworks may be an appropriate alternative or addition to resilience theories for motivating your work.

2. The Background can say more about why marriage and marital transitions, specifically, should bear on health. The background should say more about social selection and social causation as processes that account for the marriage-health association. The background can also theorize more fully whether and how BEING married or unmarried (enduring states) vs. BECOMING married or unmarried (transitions) should have distinct effects on health.

3. I would suggest some rethinking of the marital categories. Combine ‘partnered’ (which I presume to be cohabiting) with married, given that the two categories tend to show similar health outcomes for older adults. (The same is not true for younger people). Break out remarried people from continuously married people. Likewise, I don’t understand the coarse use of the ‘transition’ measure – which does not distinguish between entrances to new unions (important, given that many divorced and widowed older people do repartner) and exits. Rather than the marital status and transition measures, why not develop meaningful categories like continuously married during study period, married and then divorced in study period, etc. See work by Shinae Choi that used these more fine-grained categories effectively, although you may need somewhat coarser categories than she used b/c your analytic sample is such a small share of the overall HRS sample. For example,

Choi, S. L., & Carr, D. (2023). Older adults’ relationship trajectories and estate planning. Journal of Family and Economic Issues, 44(2), 356-372.

4. The framing says little about the importance of transition timing. For divorced people, for instance, do you know if they divorced at age 25 and then remained unmarried through old age? Or whether it was a more recent gray divorce? The same for widowed. Thinking through what marital statuses and transitions mean, and the timing thereof, may help to refine the hypotheses about their direct impacts, and the extent to which these impacts are moderated by epigenetic aging.

5. You lose a lot of cases through item-specific missing data. Why not do an imputation on the independent variable side? Or, if some of the measures are obtained only from the self-administered LBQ, use the two 50% LBQ subsamples from consecutive waves, and then pool the samples? The dropping of cases is a serious problem because you’re left with such a biased and positively selected sample. For instance, you have barely 1% who are never married, whereas in the overall sample it’s closer to 5%. If you are losing the sickest and most socially disconnected persons due to item-specific missing data, then it’s hard to interpret and contextualize your results.

6. I appreciate the gender contrast, but do not see significance tests to evaluate whether the coefficients are different in the male vs. female models.

7. In the Descriptive Statistics tables, denote statistically significant gender differences.

8. I worry that you are overcontrolling and thus are wiping out the effects of potentially meaningful variables. For instance, number of kids is highly correlated with marital status (never married tend to have <1, continuously married have more kids than those whose marriages were truncated by dissolution). Number of kids close by is correlated with total number of kids, etc. I would suggest doing a careful paring down of variables, deleting those that are the most highly confounded with your focal variables. See also Kohler, U., Class, F., & Sawert, T. (2024). Control variable selection in applied quantitative sociology: a critical review. European Sociological Review, 40(1), 173-186.

9.I have not used epigenetic clocks in my own work, so cede to the other reviewers here. At the very least, it would be instructive for readers to learn why particular measures responded differently (e.g., GrimAge and DunedeinPACE mattered for CESD). For social determinants of health (SDOH) researchers who have not used these measures, it could be helpful to understand their broader meaning and applications.

Best of luck in refining this promising project.

.

Reviewer #1: No

Reviewer #2: **Yes:** Deborah CarrDeborah CarrDeborah CarrDeborah Carr

---

## [Author Response · Author response to Decision Letter 1]

10 Feb 2026

Dear Editors and Reviewers,

I sincerely thank you for the constructive comments and helpful questions. I revised the manuscript extensively in response to all points, as detailed in this response letter.

The most important revisions include:

1. Conceptual framing: I define resilience more clearly and state the moderation hypotheses more directly, while also explaining how heterogeneity across epigenetic clocks informs interpretation.

2. Address overcontrol and reduce case loss: In response to concerns about overadjustment, I removed the child-related covariates from the main models and treated them as sensitivity checks. This change both improves the primary specification and recovers a substantial number of cases, which led to updates in several estimates.

3. Missingness and selection robustness checks: I implemented multiple imputation and inverse probability weighting sensitivity analyses (including weights for survival into the observation window and for inclusion in the DNA methylation and genotype subsamples), and I reference these checks in the revised text and supplementary tables.

I hope these revisions address the reviewers’ concerns satisfactorily, and I appreciate the opportunity to improve the manuscript.

Sincerely,

Corresponding Author

Reviewer #1:

Main concerns: My main concerns are that the paper would benefit from a more rigorous connection to the existing literature as some sentences do not have supporting references to the literature and, even more saliently, the idea of psychological resilience in epigenetic clocks has been documented in the literature (see a couple cites below). Moreover, it would be appropriate to discuss the findings in light of prior research on the association between marriage and health, which shows marriage has health benefits for men. Second, it would be useful to compare and contrast this study’s findings to those of Zhang, who also examined epigenetic clocks in HRS.

https://pmc.ncbi.nlm.nih.gov/articles/PMC10828333/

https://www.nature.com/articles/s41398-021-01735-7

Response: Thank you for these helpful suggestions. I revised the Introduction to strengthen its connection to the existing literature on resilience, epigenetic clocks, and the marriage–health association.

First, I made the discussion of psychological resilience and epigenetic aging more explicit. The manuscript already cited the Harvanek et al.’s Translational Psychiatry study linking resilience-related traits (emotion regulation and self-control) to stress-related epigenetic aging (1) and the Zhang et al.’s HRS study showing that higher psychological resilience scores are associated with slower epigenetic aging across multiple clocks (2). I expanded the text around the Harvanek citation to briefly summarize the specific pattern reported in that study and retained the Zhang citation to acknowledge related evidence in HRS, consistent with the literature highlighted in this comment.

Second, I clarified how the present study differs from prior work, including Zhang (2). Rather than treating resilience as a predictor of epigenetic aging, this study models epigenetic clocks as moderators and focuses on depressive symptoms and mortality as outcomes, testing whether epigenetic aging profiles condition responses to a later-life transition (marital status change). I also strengthened the marriage-health background in the Introduction by adding language on social selection and social causation and by noting that marriage is often especially protective for men. In the Discussion and Conclusions, I added text to interpret the gender-stratified findings in light of the marriage-health literature and prior HRS epigenetic clock research, including comparisons to Zhang.

Revised text in Introduction now reads:

Pages 4-5, lines 130-155: “Marital transitions, in particular, have been extensively examined in relation to health and longevity. Marriage is associated with better mental health, reduced risk of chronic disease, and longer survival, especially for men (3,4). In contrast, divorce and widowhood are often linked to increased psychological distress, diminished social and financial resources, and elevated mortality risk (5–10). These patterns are commonly interpreted through the dual processes of social causation and social selection. Prior work emphasizes that marriage may influence health through social support, pooled resources, and health-related regulation, while healthier and more advantaged individuals are also more likely to marry and remain married, and those in poorer health are more likely to experience marital disruption (3,4,11). Importantly, this literature distinguishes between enduring states and transitions. Being married versus unmarried can capture relatively stable differences in social integration and resources, whereas becoming divorced or widowed can operate as an acute stressor with consequences that may evolve through longer-term adaptation, with effects shaped by how recently the transition occurred and whether it involved divorce versus widowhood (12,13). Gender may further condition these associations. A social-control account argues that spousal monitoring of health behavior is one pathway through which marriage may be especially protective for men (14), consistent with evidence of gender differences in psychological adjustment and health consequences following widowhood and other marital disruptions (3,7,9).

Marital transitions are not merely psychosocial events; they also leave biological imprints. Studies have found that spousal loss is associated with physiological dysregulation, increased inflammation, and markers of accelerated aging (15–17). In the epigenetic aging literature, emerging evidence using HRS data suggests that marital status is associated with several epigenetic clocks (17). Related evidence also links partnership disruption, including divorce and widowhood, to positive DNA methylation age acceleration (18). Along similar lines, work on social loss further suggests that losing a spouse is associated with broad shifts in DNA methylation, pointing to epigenetic correlates of bereavement-related stress (19).”

Page 6, lines 179-183: “For instance, prior work shows that stress-related epigenetic aging is more pronounced among individuals with poorer emotion regulation, while stronger self-control appears to mitigate stress-related biological dysregulation, consistent with the idea that psychosocial resources can shape the biological embedding of stress (1).”

Pages 7-8, lines 202-213: “Similarly, research by Hildon et al. (20) shows that older adults often draw on a repertoire of coping strategies accumulated over the life course to manage new challenges. This perspective suggests that biological aging, as measured by epigenetic clocks, may capture not only cumulative burden but also accumulated resilience. Past studies link resilience-related characteristics to epigenetic aging, including evidence that resilience-related traits can buffer stress-related epigenetic aging (1) and that psychological resilience is associated with epigenetic clocks in HRS (2). Building on this literature, we use epigenetic clocks as moderators to test whether epigenetic aging profiles condition responses to a later-life transition, with depressive symptoms and mortality as the outcomes. In keeping with the process-oriented definition above, resilience in this study is assessed as reduced downstream consequences following marital status change, rather than requiring the complete absence of any adverse response.”

Revised text in Discussion and Conclusions (pages 30-31, lines 568-609): “This interpretation is supported by theoretical frameworks on resilience, which emphasize the interplay of psychological, social, and biological factors in adaptive functioning (21–23). Resilience develops not only from personal characteristics such as temperament or emotional stability but also from life experiences that foster coping strategies and emotional regulation. In addition to these psychological resources, researchers increasingly recognize a biological dimension of resilience, seen in the body’s capacity to adapt to stress at the physiological level. Findings from recent research (1,2,24) indicate that resilience-related traits such as self-control and emotion regulation can buffer the negative impact of cumulative stress on epigenetic aging. Related reviews further link trauma, discrimination, and chronic adversity to accelerated epigenetic aging, while also emphasizing that psychosocial resources shape biological embedding across the life course (24–26). At the same time, resilience processes may not manifest uniformly across all epigenetic clock constructs or health outcomes, given differences in what clocks capture and how outcomes reflect downstream risk (25,27,28). In this light, even if evidence of moderation is not uniform across outcomes or clocks, the epigenetic moderation results in this study are consistent with the idea that resilience can be biologically embedded. Accumulated life experiences may strengthen the body’s capacity to adapt to and withstand future adversities.

Importantly, our approach differs from prior research on resilience and epigenetic aging by treating epigenetic clocks as potential effect modifiers rather than as outcomes. Whereas psychological resilience has been modeled as a predictor of epigenetic aging in Zhang et al.’s HRS analysis (2), and resilience-related traits have been shown to moderate stress-related epigenetic aging (1), we test whether epigenetic aging profiles condition responses to a later-life transition, recent marital status change, using CESD score and mortality as outcomes. These approaches address different parts of the same resilience process. In Zhang et al. (2), higher psychological resilience was associated with slower epigenetic aging, which can be interpreted as lower biological vulnerability. In our analyses, epigenetic aging profiles are used to stratify vulnerability and adaptation, and we find that moderation is more apparent for mortality than for CESD score. In particular, Zhang et al. found that resilience was most robustly associated with slower epigenetic aging for clocks including Levine, Zhang, Garagnani, and GrimAge (2), whereas in our analyses moderation was clearest in the mortality models and concentrated in a smaller subset of clocks, with Hannum, Garagnani, and GrimAge showing the strongest evidence. This outcome contrast is plausible given that several clocks in common use were developed to capture healthspan or mortality risk (e.g., GrimAge and Levine/PhenoAge) or the pace of aging (DunedinPACE), and these measures are often more tightly linked to survival outcomes than clocks trained primarily on chronological age (25–29).

Analyses using gender-stratified samples revealed important gender differences in responses to marital status change and in the moderating effects of epigenetic clocks. Overall, marital status change and its interactions with epigenetic clocks were more consistently evident in men’s models than in women’s models. These findings are consistent with prior research showing that men may suffer more immediate emotional consequences following marital dissolution (30,31), and with gendered accounts of marriage as a source of social regulation and support that may be especially consequential for men (12,14). Specifically, marital status change was associated with higher CESD scores more clearly among men than women. In addition, the clearest evidence of moderation appeared in men’s mortality models, most notably for Hannum and GrimAge, with Zhang also showing moderation among men.”

Specific Text comments:

#1 PAGE 3 Check grammatical structure of this sentence.

For instance, using data from the National Longitudinal Study of Adolescent to Adult Health (Add Health), (16) showed that educational attainment, income, health behaviors, and stress exposure robustly predict variation in epigenetic age acceleration across multiple clocks.

Response: Thank you for catching this. The line is now revised to: “For instance, analyses using data from the National Longitudinal Study of Adolescent to Adult Health (Add Health) showed that educational attainment, income, health behaviors, and stress exposure robustly predict variation in epigenetic age acceleration across multiple clocks (32).”

#2 PAGE 4

“Although aging is often portrayed as a period of vulnerability,”

This does not read quite right. Given aging is an action, it’s not a life stage. Perhaps the author means that older age is a period of vulnerability or that aging heightens biological vulnerability to dysfunction.

Response: Thank you for pointing this out. I revised the sentence for conceptual precision by referring to older age as a life stage rather than “aging” as a process. The sentence now reads: “Although older age is often portrayed as a period of vulnerability,”

#3 RESULTS SECTION

The results section needs to report estimates in text, using a reporting style such as AMA or APA.

Response: Thank you. I revised the Results section to report key model estimates in the text using APA-style reporting (e.g., b, SE, and p values), with full model results retained in the main text and the corresponding supplementary tables.

#4 PAGE 25

“ Age may not be as neutral a concept as it initially appears.”

Please revise or better support the comment. Aging is often considered a negative effect and associated with negative stereotypes in US contexts.

Response: Thank you for this suggestion. I revised the opening of the Discussion and Conclusions to better support the point and to clarify what I mean by “not neutral,” emphasizing that chronological age is often used as an administrative marker but is also shaped by social meanings, including negative age stereotypes in the U.S. context, and that lived experience across the life course can become biologically embedded. The sentence now reads: “Chronological age is often treated as a neutral marker, yet it carries social meanings and reflects lived experience that can become biologically embedded over the life course (33–36).”

#5 PAGE 25

The two passages pasted below raise two issues. First, the conceptualization of the clocks is positioned as both biological (second passage) and ‘more than biological’ (first passage). The position ought to be consistent throughout the paper and whichever position is used it ought to be well reasoned and supported by the extant literature. Second, the author presents a useful description of the clocks and their constructure in the methods but in the first passage appears to dismiss that there are differences in the clocks and that half showed significant associations and half did not. These points ought not to be dismissed but rather discussed. Moreover, specifically how the findings support the assertion that the clocks reflect social and psychological effects needs to be better articulated.

Although significant associations were observed for fewer than half of the epigenetic clocks analyzed, the findings suggest that epigenetic aging reflects more than simple biological deterioration. Rather, epigenetic clocks may serve as integrated markers of biological, social, and psychological aging, capturing cumulative life experiences that could include adaptive responses to adversity.

VERSUS

PAGE 28

In this light, the epigenetic moderation effects found in this study further validate the idea that resilience is biologically embedded, with accumulated life experiences strengthening the body's capacity to withstand future adversities.

Response: Thank you for this thoughtful comment. I revised the Discussion and Conclusions to keep the conceptualization of epigenetic clocks consistent across the manuscript and to discuss the heterogeneity across clock constructs directly, rather than treating it as incidental. First, I clarify that epigenetic clocks are DNA methylation-based biological measures, while also noting that DNA methylation is responsive to socially patterned exposures and behaviors, so variation across clocks can reflect biological embedding processes (25,27,32,37,38). Second, I explicitly interpret the fact that only a subset of clocks shows significant associations as informat

---

## [Decision Letter · Decision Letter 1]

5 Apr 2026

Epigenetic Clocks Moderate the Impact of Marital Status Transitions on Health in Older Adults

PONE-D-25-30828R1

Dear Dr. Lin,

We’re pleased to inform you that your manuscript has been judged scientifically suitable for publication and will be formally accepted for publication once it meets all outstanding technical requirements.

Kind regards,

Vincenzo De Luca

Academic Editor

PLOS One

Additional Editor Comments (optional):

Reviewers' comments:

Reviewer's Responses to Questions

**Comments to the Author**

Reviewer #1: All comments have been addressed

Reviewer #2: All comments have been addressed

2. Is the manuscript technically sound, and do the data support the conclusions?

Reviewer #1: Yes

Reviewer #2: Yes

3. Has the statistical analysis been performed appropriately and rigorously?

Reviewer #1: Yes

Reviewer #2: Yes

4. Have the authors made all data underlying the findings in their manuscript fully available?

Reviewer #1: Yes

Reviewer #2: Yes

5. Is the manuscript presented in an intelligible fashion and written in standard English?

Reviewer #1: Yes

Reviewer #2: Yes

Reviewer #1: Using data from 1,449 non-Hispanic White participants in the Health and Retirement Study, the study analyzed thirteen epigenetic clocks to assess associations between marital transitions, depressive symptoms, and mortality, adjusting for genetic and social factors. Interaction terms tested whether epigenetic clocks moderated these associations. The findings support the notion that epigenetic clocks capture effects of accumulated life experiences, which include those that reflect deterioration and resilience. The author proposes that the current study expands the literature on epigenetic clocks to include psychosocial adaptation or resilience.

The topic is of interest, the paper is well written and the analyses are well done. The author provided an excellent response to the concerns I raised in my initial review. This paper would make a valuable contribution to the existing literature.

I thank the author for a thorough and thoughtful response to my prior concerns.

Reviewer #2: The authors are to be commended for their thorough and responsive revision. The study is now better situated within relevant literatures, concepts have been clarified, and results presented clearly and concisely. I have no further suggestions for revision.

.

Reviewer #1: No

Reviewer #2: **Yes:** Deborah S CarrDeborah S CarrDeborah S CarrDeborah S Carr

---

## [Editor Report · Acceptance letter]

PONE-D-25-30828R1

PLOS One

Dear Dr. Lin,

I'm pleased to inform you that your manuscript has been deemed suitable for publication in PLOS One. Congratulations! Your manuscript is now being handed over to our production team.

Kind regards,

on behalf of

Dr. Vincenzo De Luca

Academic Editor

PLOS One